# Temporal shift of groundwater fauna in South-West Germany

Fabien Koch[1], Philipp Blum[1], Heide Stein[2], Andreas Fuchs[2], Hans Jürgen Hahn[2], Kathrin Menberg[1]

[1]Institue of Applied Geoscience (AGW), Karlsruhe Institute of Technology (KIT), Kaiserstraße 12, 76131 Karlsruhe, Germany
[2]Institute for Environmental Sciences, University-Kaiserslautern-Landau (RPTU), Fortstraße 7, 76829 Landau, Germany

5    *Correspondence to*: Fabien Koch (Fabien.koch@kit.edu)

**Abstract**

Groundwater is an important source of freshwater, drinking water and service water for irrigation, industrial and geothermal uses and the largest terrestrial freshwater biome of the world. In many areas, this habitat is naturally or anthropogenically threatened. This study uses long-term groundwater data from South-West Germany to identify shifts in groundwater fauna due to natural or anthropogenic impacts. Comprehensive analysis of metazoan groundwater fauna and abiotic parameters from 16 monitoring wells over two decades revealed no overall temporal trends for fauna abundance, biodiversity in terms of number of species, as well as no significant large-scale trends in abiotic parameters. While nine wells out of 16 show stable ecological and hydro-chemical conditions at a local level, the remaining wells exhibit shifting or fluctuating faunal parameters. At some locations, these temporal changes are linked to natural causes, such as decreasing dissolved oxygen contents or fluctuating temperatures. A multivariate PHATE (Potential of Heat-diffusion for Affinity-based Trajectory Embedding)-analysis suggests that, beside the hydrogeological setting, varying contents of sediment and detritus impact faunal abundance. By examining aerial images of the surroundings of individual wells, we found that anthropogenic impacts, such as construction sites and surface sealing, can cause significant shifts in groundwater fauna and changes in the ecological status in positive as well as negative directions. However, variable faunal composition and abundances were also observed for sites with very stable abiotic conditions in anthropogenically less affected areas such as the Black Forest. These findings indicate that hydro(geo)logical changes and surface conditions, such as land use, should be assessed in line with hydro-chemical parameters to better understand changes in groundwater fauna. Accordingly, reference sites for natural conditions in ecological assessment and biomonitoring schemes for groundwater protection should be selected carefully.

**Keywords**

25   Groundwater ecology; Groundwater biodiversity; Groundwater fauna; Groundwater ecosystem; Stygofauna; Biomonitoring

## 1. Introduction

Groundwater is an important source of freshwater, drinking water and service water for irrigation, geothermal and industrial uses (Job, 2022; Siebert et al., 2010; Stauffer et al., 2013). Furthermore, groundwater ecosystems build the largest terrestrial freshwater biome of the world (Griebler et al., 2014a). This habitat is considered species-rich (>100,000 species) with many endemic taxa (Culver and Holsinger, 1992).

However, human activities and natural changes threaten this habitat in many areas (Becher et al. 2022), put pressure on its ecosystem and groundwater communities, and alter general biogeochemical processes (Griebler et al., 2016). Changes in water quality and water volume are driven by natural processes (Goater, 2009), yet groundwater abstraction for irrigation, drinking water and mining activities changes groundwater volume and, therefore, groundwater levels (Danielopol et al., 2003; Hancock et al., 2005). Changes in groundwater quality can also be caused by pollutants originating from agriculture (Korbel et al., 2022; Di Lorenzo et al., 2020; Di Lorenzo and Galassi, 2013), urbanisation (Hallam et al., 2008), surface waters (Danielopol et al., 2003a; Kristensen et al., 2018), heavy industry (Hose et al., 2014) and thermal pollution (Menberg et al., 2013; Taylor and Stefan, 2009a; Tissen et al., 2019; Zhu et al., 2010). Moreover, climate change, which increases groundwater temperatures (Figura et al., 2011a; Menberg et al., 2014a; Tissen et al., 2019), puts pressure on groundwater and its ecosystems. Those environmental pressures can alter groundwater fauna community structures (Korbel et al., 2022) and, therefore, induce changes in groundwater biodiversity and species dominance (Goater, 2009). Typical implications of this are changes in the natural variation of population cycles, shifts in the composition of the community as ubiquitous surface-water species can outcompete and replace groundwater species, and, finally, species extinction (Danielopol et al., 2003). The potential decline or even loss of groundwater communities compromises the functioning of an aquifer and its ecosystem, resulting in deterioration of groundwater health (Hancock, 2002; Hancock et al., 2005).

To assess groundwater's health or ecological status, schemes for monitoring this ecosystem, i.e., biomonitoring, become increasingly important, as already implemented for surface water (Haase et al., 2023). Biomonitoring is defined as the 'use of biological systems (organisms and organism communities) to monitor environmental change over space and/or time' (DIN EN 16413: VDI, 2018). More specifically, by repeatedly or permanently monitoring organisms or organism communities, the quality of a habitat environment is recorded and assessed, and the state of an ecosystem and its dynamics in time and space are evaluated (Mösslacher and Notenboom, 1999; Underwood, 1997). For application in groundwater, two strategies are currently available (Friberg et al., 2011; Mösslacher et al., 2001; Mösslacher and Notenboom, 1999). In 'active' biomonitoring, standardised organisms with a known origin from the laboratory are inserted in groundwater, and their behaviour is monitored in this habitat (see the biological early warning system of the GroundCare project Spengler et al. (2017)). In contrast, in 'passive' biomonitoring, bioindicators (wildlife population) are sampled from their natural habitat and their behaviour is examined in the laboratory (Brielmann et al. (2011) or as in most studies, the faunal communities found are analysed to conclude the prevailing conditions (Fuchs, Hahn, and Barufke 2006; Hahn and Fuchs 2009; Korbel and Hose 2015; Stein et al. 2010; VDI 16413 2018).

The advantage of biomonitoring compared to monitoring abiotic parameters is that organisms and communities act as remote sensors integrating environmental conditions and stress over their lifetime, undergo quantitative and qualitative alterations and thus can provide information on medium- to long-term environmental conditions and changes (Conti, 2008). Hence, for most existing biomonitoring programs, faunal community composition, different species and taxa, and an abundance of other taxonomic groups are considered (Conti, 2008; Friberg et al., 2011). Macroinvertebrates are often used for biomonitoring as their taxonomy is well known; they are sensitive to many stressors and can be sampled easily and repeatedly over specific time periods, which results in more accurate indications of diversity, richness and composition among all groundwater biota (Conti, 2008; Friberg et al., 2011; Lennon, 2019).

Surface waters have been a key focus of aquatic research due to their accessibility and visibility. The assessment of surface waters is typically based on biological, hydro-morphological and physico-chemical criteria and is defined in detail in the European Water Framework Directive (WFD). Accordingly, there is a large number of studies with long-term data on aquatic surface ecosystems. In this context, a recent study collected 1,816 time series from riverine systems in 22 European countries from 1968 to 2020 (714,698 observations) to investigate freshwater biodiversity (Haase et al., 2023). The authors conclude that standardised, long-term and large-scaled monitoring can be used to effectively characterise temporal changes in biodiversity and environmental drivers and identify sites at high risk.

A prime example of long-term monitoring in the field is the Swedish national surface water monitoring program, which began with the first research on the Mälaren lake in 1964 aiming to better understand the eutrophication. This project contributed significantly to understand the effects of climate change, land use and post-glacial rebound on water quality. Today, the combined program comprises monitoring of water chemistry and biodiversity in 114 streams and 110 lakes, and a probability-sampling program includes 4800 lakes (Fölster et al., 2014). With the advent of the WFD, reference sites from Swedish monitoring were used for inter-calibration of northern continental Europe. Data of this monitoring program 'play a key role in past and present national and international environmental commitments including Swedish environmental objectives, critical load assessments, and many aspects of EU legislation, such as the WFD, Habitats Directive, and Nitrate Vulnerable Zone Directive' (Fölster et al., 2014).

However, until now, a lack of repeated and long-term samplings of groundwater fauna has limited the understanding of the basic biology and ecology of many species, and there is a notable absence of analysis of temporal evolution and variation of groundwater fauna (Koch et al., 2022). Repeated samplings over multiple years can be found in very few locations, mainly in Central Europe and Australia (Goater, 2009; Korbel and Hose, 2017; Marmonier et al., 2000; Menció et al., 2014). Goater (2009) analysed groundwater fauna data from an 8-year monitoring program (1999-2007) at 21 wells in Australia, observing changes in stygofauna species presence, population numbers and community assemblages, as well as a shift from an amphipod-dominated to a copepod-dominated system. A second study in Australia conducted four sampling campaigns between 2007 and 2010 at 20 wells to investigate stream–aquifer relationship (Menció et al., 2014). Further ecological research focused on examining environmental and human influences on the distribution of biota and groundwater ecosystem health. A study in New South Wales at 15 sites with six samplings revealed only minor variation in ecological conditions between 2007 and 2015

(Korbel and Hose, 2017). Moreover, ecosystem health benchmarks are associated with aquifer typology rather than applied only to local areas. In Europe, Marmonier et al. (2000) also revealed limited temporal variations in the biodiversity of the interstitial fauna of artificial aquatic systems for three successive years (1995-1997). Long-term data on groundwater ecology and chemistry is available across the state of Baden-Württemberg, Germany, from a continuous monitoring program of 44 sites sampled annually or bi-annually between 2002 and 2022 (Fuchs, 2007; Fuchs et al., 2006; Stein et al., 2015), which makes the state of Baden-Württemberg (BW) one of the most densely investigated areas worldwide concerning groundwater fauna with 2026 samplings at 950 sites (Koch et al., 2022).

The main objective of this study is to identify changes in groundwater fauna in the German state of Baden-Württemberg over the last decades on both regional and local scales. We hypothesize that these changes can be related to natural or anthropogenic stress observable through changes in abiotic parameters, such as temperature increase from climate change and high nitrate concentration from agricultural fertilisation, as well as land use changes, such as an increase in surface sealing. To this end, multiple abiotic and biotic groundwater parameters of all wells are first jointly analysed over time, before individual sites are scrutinised for small-scale changes. Furthermore, changes in groundwater ecosystems on different spatial scales and the implications of the observed changes for biomonitoring are assessed. Finally, we identify ecological and physico-chemical parameters most suitable for robust biomonitoring.

## 2. Material and method

The following workflow was developed to address our objective (Figure 1). The available groundwater data of the study site (i.e. the state of Baden-Württemberg) was reviewed, and observation wells for additional sampling in 2020 were selected (Figure 1, step 1, site selection). Afterwards, the biotic and abiotic data was temporally and statistically analysed on different spatial scales (local to state-wide, Figure 1, steps 2 and 3, temporal and statistical analyses). Finally, three individual wells were analysed in detail concerning changes in land use and abiotic parameters (Figure 1, step 4, local scale analysis).

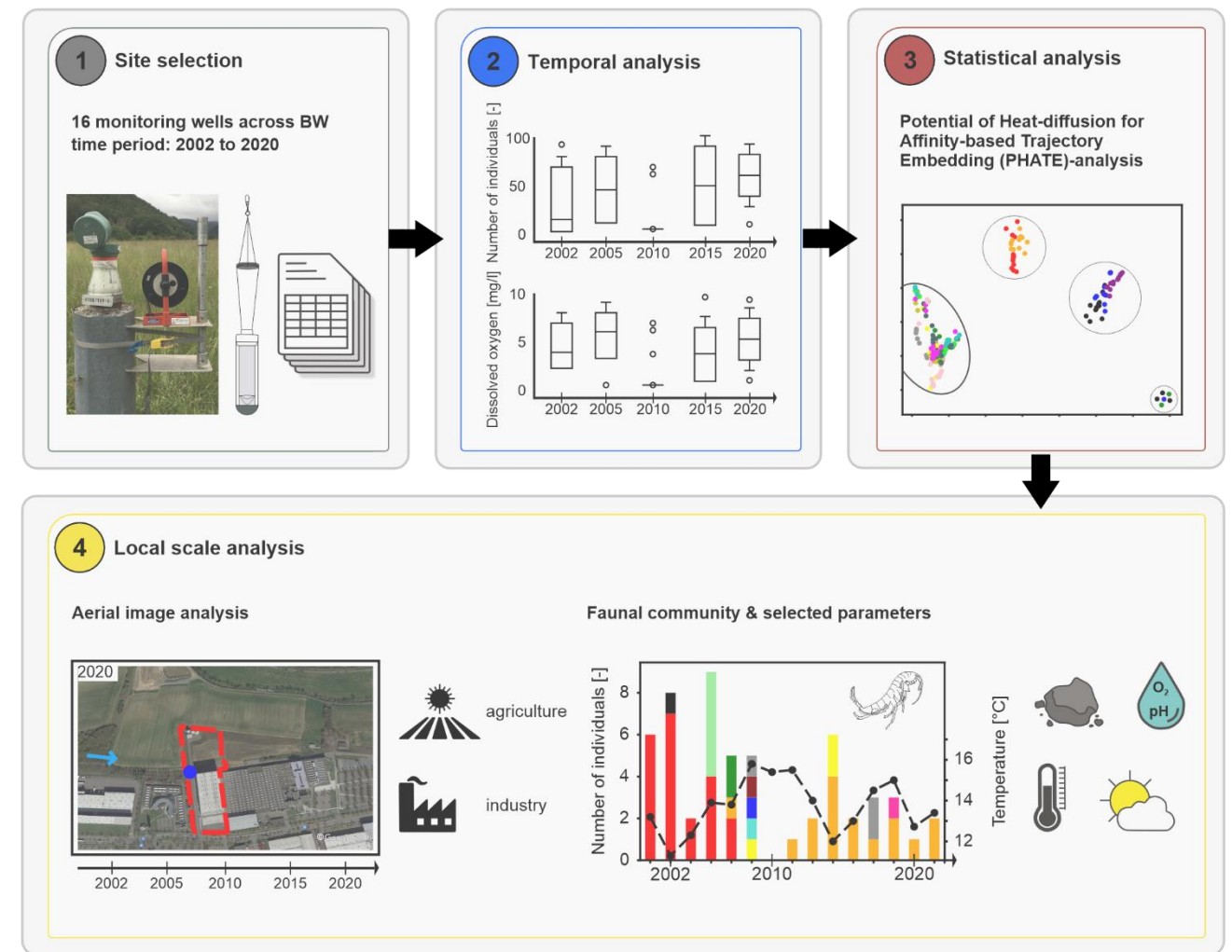


**Figure 1: Developed workflow including four steps: (1) site selection and database (image source: Fabien Koch, KIT)., (2) temporal analysis, (3) statistical analysis and (4) local scale analysis (image source: Google Earth Pro** (Google LLC., 2022))**.**

## 2.1 Site selection

The State Office of Environment, Measurements and Nature Conservation (Landesanstalt für Umwelt, Messungen und

Naturschutz Baden-Württemberg, LUBW) maintains an extensive network of up to 2,600 groundwater observation wells (Landesanstalt für Umwelt Messungen und Naturschutz Baden-Württemberg, 2013). In addition, the locations of the groundwater level monitoring network and the database of all monitoring sites in Baden-Württemberg are also considered, which results in 50,000 initial sites. Of these, 304 wells were analysed faunistically at least twice in the past (Fuchs, 2007) (Figure 2). The 304 sites were selected based on a representative distribution within the different natural areas and aquifer

types in the state of Baden-Württemberg, as well as accessibility and absence of installed measurement devices, pumps, etc. within the wells (Fuchs, 2007). Based on the initial measurement results in 2001/2002, out of the 304 sites, 44 were selected

for annually or bi-annually sampling between 2006 and 2022 (Stein et al., 2015). Faunal colonisation, a good area- and aquifer-type coverage and availability of physico-chemical measurements were considered for this selection. Out of these 44, 16 wells were selected for the current study based on spatial coverage of the study area, aquifer type, land use type, well depth, faunal colonisation during the past two decades, availability of time series of physico-chemical parameters, and an average content of dissolved oxygen higher than 1 mg/l, which is a limiting factor for faunal colonisation (Griebler et al., 2014b).

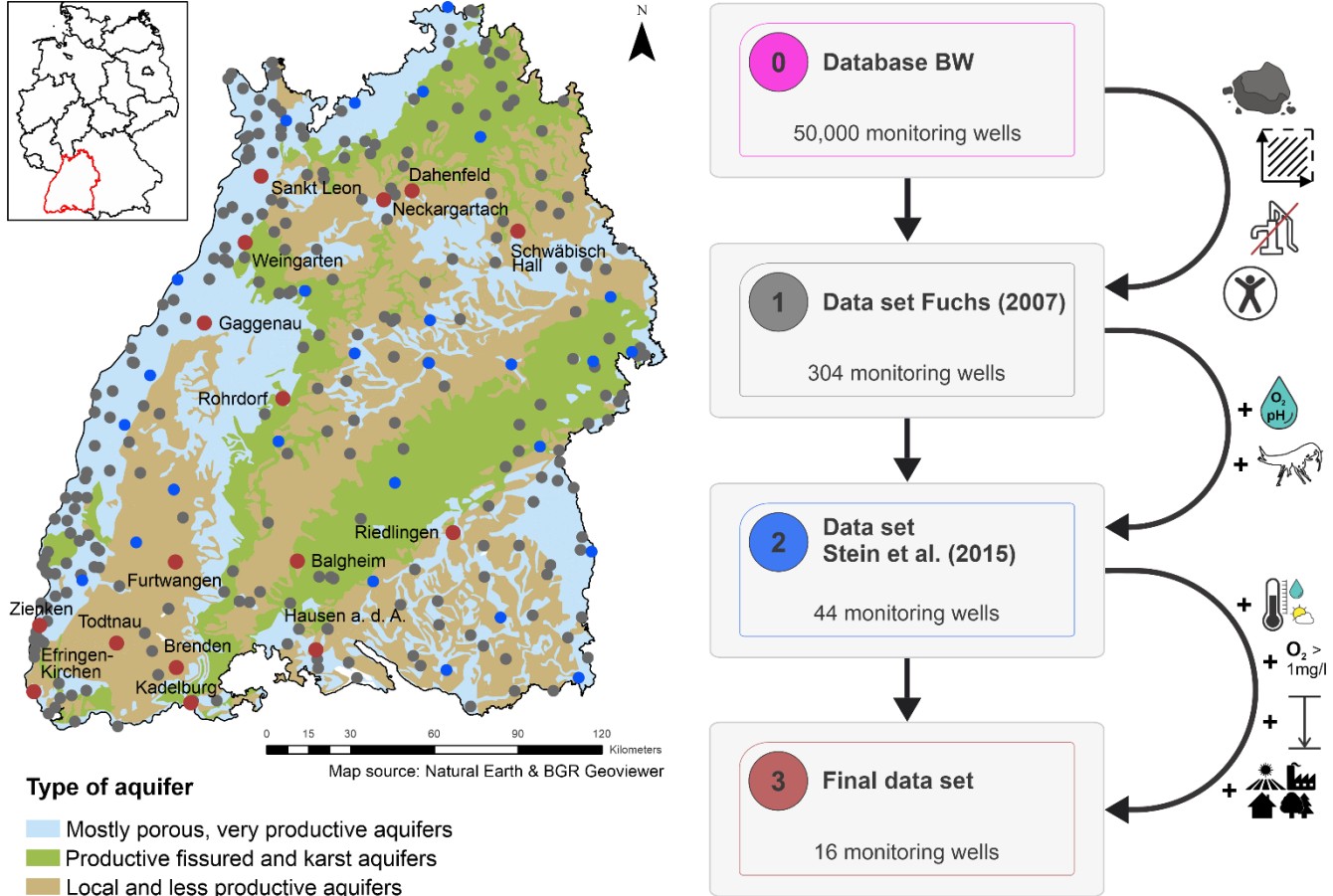

**Figure 2: Study area, the state of Baden-Württemberg in South-West Germany, including the aquifer types according to the classification of the Federal Institute for Geosciences and Natural Resources (left corner) and selected monitoring wells according to the selection process on the right** (Hydrogeologie von Deutschland 1:1.000.000 – Klassifikation gemäß der Standardlegende für Hydrogeologische Karten (HY1000–SLHyM), 2022)**.**

## 2.2 Groundwater sampling

Faunal groundwater sampling in the 16 selected observation wells took place twice in 2001/2002 (for analysis in this study, only data from June-September was used), annually from 2006 to 2014, and then bi-annually in August/September until 2020. At the beginning of each sampling, the depth of the groundwater table and of the well were measured using an electrical contact gauge. Afterwards, water standing in the well (i.e. well water) was taken with a bailer (750 ml Aquasampler of the Bürkle

GmbH, Lörrach) from the bottom of the well. In these samples, oxygen concentrations (Intellical LD0101, luminescence-based optical probe; accuracy: ± 0.1 mg/l), carbonate hardness, pH (PHC101; accuracy: ± 0.002) and electrical conductivity (TDS-measuring instrument Meditech/TenYua) were measured using an HQ40D portable 2-channel multimeter (Hach Lang GmbH,

2022). Additional samples were taken to determine the amount of sediment and further chemical analyses (content of dissolved carbon, organic nitrogen, phosphate, ammonium, ortho-phosphate, total phosphate, total bacterial count and total number of cells).

The faunal sampling was conducted using a modified Cvetkov net as described by Hahn and Fuchs (2009). More specifically, a plankton net consisting of a gauze funnel with a mesh size of 73 μm attached to a collection vessel (50 ml centrifuge tube)

with a weight was used (Figure 1, step 1). With the help of a fishing rod or a winch for larger depths, the net sampler was lowered into the well down to the bottom. To sample as much fauna as possible, the net sampler was quickly raised about 1.5 m and lowered again ten times. The collected samples were stored in a cooling box at about 8 °C until fixation with 96 % ethanol on the same day. The dye Rose Bengal ($C_{20}H_2Cl_4L_4Na_2O_5$ by Thermo Scientific Chemicals), is added until 2015 and Eosin B ($C_{20}H_6Br_2N_2Na_2O_9$) from 2015 onwards, which colours organic matter in a pink hue for easier determination of

groundwater fauna. Faunal samples were sorted on order level and determined on species level under a magnification between 10 and 20. Based on this determination, further faunal parameters were calculated, such as the total abundance, number of taxa or species, classification into stygofaunal classes (stygoxenic, stygophil, stygobiont) and ecological status according to Griebler, Stein, et al. (2014). Limitations regarding the sampling method must be considered when interpreting the faunal results. With the help of a net sampler, only the standing water of a groundwater monitoring well is sampled. This water is

affected by filter effects, which can lead to an overrepresentation of larger individuals, such as amphipods (Hahn and Matzke, 2005; Korbel et al., 2017). Larger, more active and sessile organisms, on the other hand, may not be fully captured by sampling using pumps, as they are subject to filter effects, especially in fine sediments. However, studies have shown that the proportion of species and presence/absence of taxa is similar between the two sampling methods (Hahn and Gutjahr, 2014; Hahn and Matzke, 2005; Korbel et al., 2017).

Hydro-chemical groundwater sampling from pumped aquifer water was performed by the LUBW, with measurement results being provided in an annual catalogue, which is publicly accessible online (Jahresdatenkatalog Grundwasser, 2022). Of all 144 available parameters, 42 are used in this study, in particular standard cations and anions, heavy metals and inorganic trace substances, pesticides and aromatic hydrocarbons (Table S1). Parameters with a lack of temporal resolution are excluded. Aquifer water is analysed in-situ in the field with probes in accordance with the LUBW groundwater sampling guidelines and

in line with the applicable DIN standards for electric conductivity (DIN EN 27888), water temperature (DIN 38404 4), pH-value (DIN EN ISO 10523), oxygen (DIN EN 25814, in-situ or using a flow-through system) and base capacity (DIN 38409-7) (Landesanstalt für Umwelt Messungen und Naturschutz Baden-Württemberg, 2013). Laboratory measurements are conducted in an accredited laboratory according to DIN EN ISO/IEC 17025. Annual air temperature data was taken from the German Meteorological Service  (Raster der Monatsmittel der Lufttemperaturminima (2m) für Deutschland, Version v19.3,

175  2022).

### 2.3 Statistical analysis

To better understand large-scale spatial relationships and the structure of the high-dimensional groundwater data, a PHATE (Potential of Heat-diffusion for Affinity-based Trajectory Embedding)-analysis is conducted (Moon et al., 2019). This method has shown to provide meaningful results for handling financial data (Grzybowska and Karwański, 2022), prediction of disease outbreaks (Kuchroo et al., 2022), learning of brain activation manifolds (Busch et al., 2022), RNA sequencing (Moon et al., 2019) and investigation of groundwater ecosystems (Koch et al., 2021). In contrast to most existing methods, a PHATE-analysis allows to obtain biologically interesting structures from increasingly large, modern datasets while accounting for the high level of noise inherent in biological datasets.

PHATE is a dimensionality reduction method that generates a low-dimensional embedding specific for visualisation, in the form of scatter maps. Thus, an accurate, denoised representation of a data set's global and local structures is provided without imposing strong assumptions on the design of the data. The PHATE algorithm computes the pairwise distances from the data matrix. It transforms the distances to affinities to encode local information by applying a Gaussian kernel function to Euclidian distances. Afterwards, the local affinity matrix is row-normalised to get an N x N stochastic matrix. Thus, the normalized kernels form a probability transition matrix. Using diffusion processes, global relationships are learnt and encoded using the potential distance. In detail, the von-Neuman entropy is used to determine a diffusion time scale. Markov processes are then used to define the distance between the points in t-steps. In the next step, a random walk is performed and fixed initial conditions are used for the data generation process. The resulting distance is then used for the final step of the algorithm (for more details see Grzybowska and Karwański (2022)). Metric multidimensional scaling (MDS) embeds the likely distance information into low dimensions for visualisation (Moon et al., 2019). Thus, objects with similar characteristics are close to each other in the final graph. The axis labels PHATE1 and PHATE2 are dimensionless and represent parameter combinations, i.e. distinct axes in the multidimensional parameter space (comparable to similar analyses, e.g. MDS). Another advantage of this method is that biological and abiotic parameters can be considered together, allowing identification of further dependency structures. In this study, the analysis is conducted using 15 physical, biotical and (hydro-)geological input parameters (Table S2). The selection of parameters is based on previous studies, as well as empirical values on influencing factors from the literature (Hahn, 2006; Koch et al., 2021; Korbel et al., 2018; Stein et al., 2012). The PHATE-analysis is carried out using a Python code provided by Krishnaswamy Lab on GitHub (see Moon et al. (2019)). From each well, data of all annual measurement campaigns are analysed separately to assess temporal variations in faunal communities and abiotic conditions.

## 3.   Results

### 3.1  Faunistic overview

Considering the entire investigation period (2002 - 2020), the faunal colonisation differs across the study area, with between 52 and 1800 individuals and up to 42 different species per well, respectively (Figure 3). The highest abundances in the state

of Baden-Württemberg (BW) were found in the porous aquifers in the northeastern part of BW and in the southern Upper Rhine Valley, while the lowest abundances were found in fractured and karst aquifers, as well as in less productive aquifers (Figure 2). The spatial distribution of the faunal community according to taxonomic groups is shown in Figure 3b. In the southeast of the state, Crustaceans (mainly Cyclopoids) dominate. In karst aquifers of the Swabian Alb and the local and less productive fissured aquifers of the Southern Black Forest, Amphipods are more frequent (e.g., Balgheim, Todtnau, Brenden). Wells in the northern Black Forest and along the Rhine in more productive aquifers are additionally colonised by Isopods, Harpacticoids, Nematods, Ostracodes and/or Annelids. Synacrids were found only in wells in the northern part of BW (except for Kadelburg), in the north of Upper Rhine Valley and the catchment of the Neckar River.

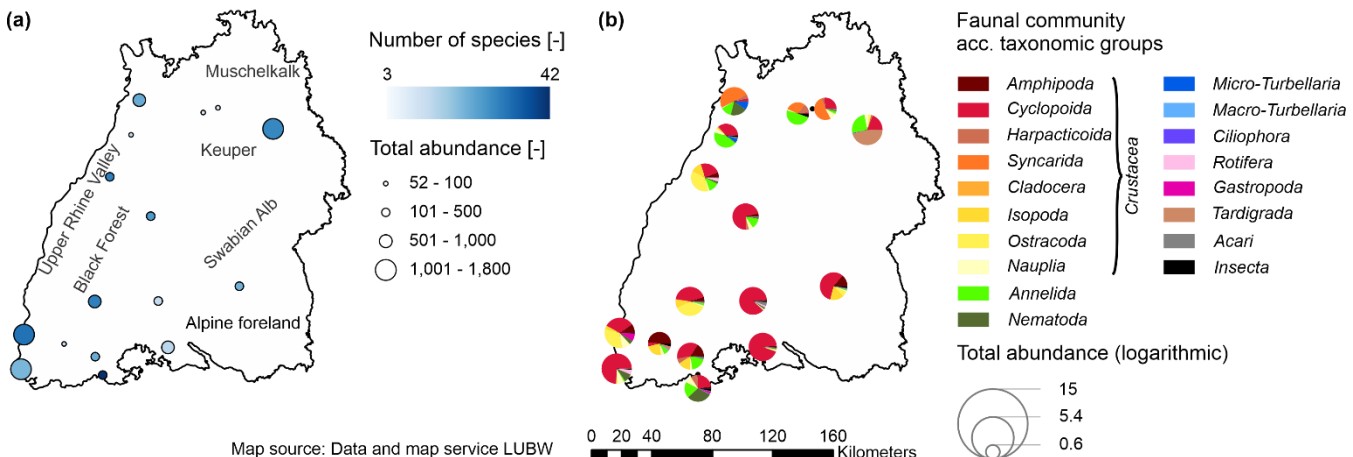

**Figure 3: Maps of (a) the number of species and the total abundance and (b) the faunal community according to taxonomic groups as a sum of all measurements of all years (2002 – 2020) for each of the 16 monitoring wells in the study area.**

## 3.2 Temporal analysis

From 2002 to 2020, the number of individuals of all 16 monitoring wells ranged between 10 and 50 on average per year, with no observable temporal change in the average total abundance or its variation (Figure 4a). The same applies to the number of species (Figure 4b). Also, no significant changes are revealed over time for the abiotic parameters (Figure S1). The year 2007 stands out with a high number of individuals and species (Figure 4), as well as a low share of stygobiont species. This year also shows a higher content of dissolved oxygen and electrical conductivity (Figure S1). This is particularly true for wells in Efringen, Rohrdorf, Zienken, Schwäbisch Hall, and Gaggenau.

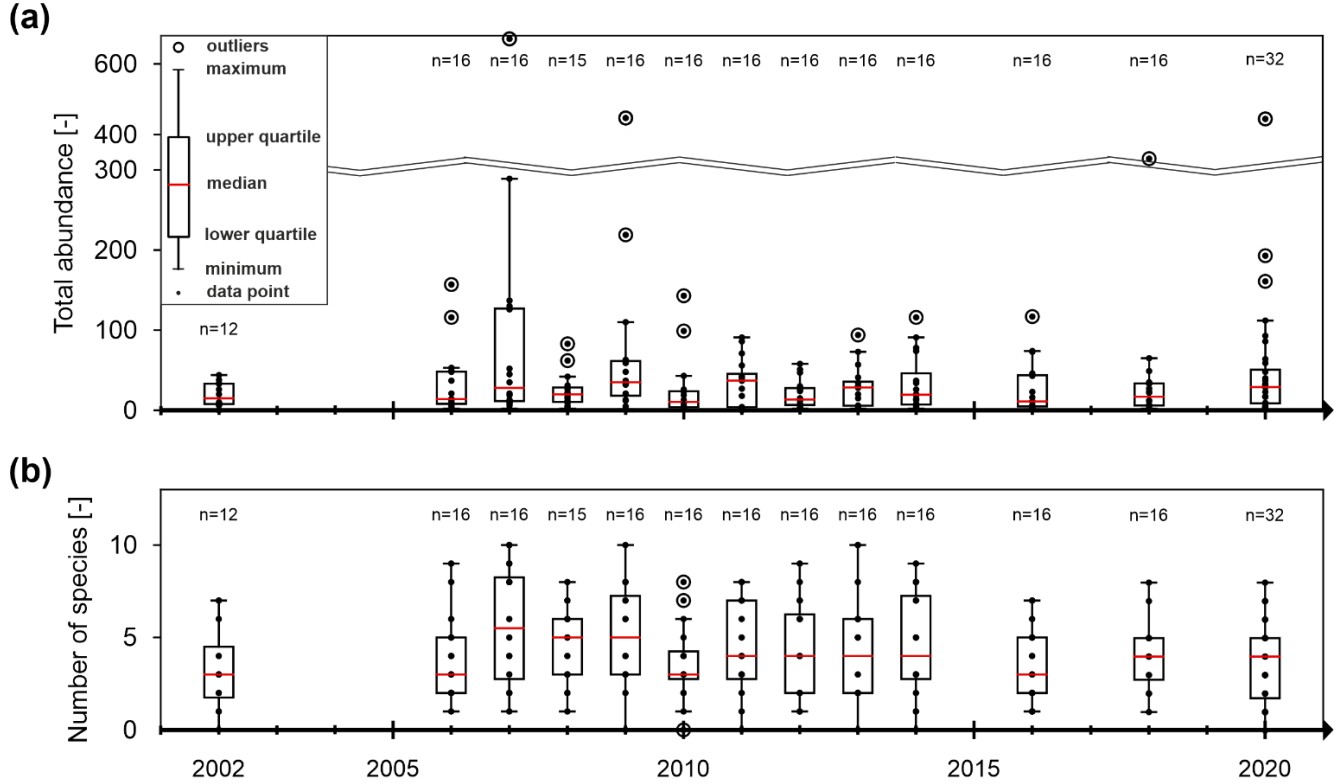

**Figure 4: Boxplots of important biotic parameters between 2002 and 2020 of all 16 wells combined: (a) total abundance and (b) number of sampled species. The dots represent all available data points of each year. "n" indicates the number of measuring points. No sampling was conducted in years with no boxplot.**

While there is no apparent trend on the regional scale, there are significant variations between individual years for some individual wells (Figure 5). Regarding the faunal parameters, no single well exhibits a significant positive development concerning faunal abundance or biodiversity. Only the well in Schwäbisch Hall shows a slightly increasing trend in abundance over the last five years. This well is characterised by the presence of Tardigrads (Figure **3**, light brown colour and in the online dataset (Koch et al., 2024)), which are typical for small surface water bodies, and wells with organic material from the surface (leaves, moss, etc.) (Schminke et al., 2007). Moreover, this well shows a gradual change in the faunal community from one single stygobiont species to multiple stygobiont and stygophile species. On the other hand, there is a clear decrease in the abundance and faunal diversity in Todtnau und Zienken (Figure 5a &b), which in the case of Zienken is linked to decreasing dissolved oxygen contents with < 1 mg/l in 2014 (Figure 5e). In Zienken periods with higher abundances from 2007 to 2009 and 2013 to 2014 coincided with higher numbers of ubiquitous species from the surface and a higher bacterial count. Moreover, fluctuations of abundance and number of species are present in several wells, e.g. in Gaggenau and Efringen (Figure 5a and 5b). Besides four stygobiont indicator species in Gaggenau, all other species only appear sporadically (Figure 5b). Also, the increase in oxygen concentration after 2006 in Gaggenau is striking, as well as the higher number of Oligochaetes and Nematodes (Figure 5c).

An important parameter for the ecosystem status is the ratio of Crustaceans and Oligochaetes, which serves as a basis of the ecological assessment scheme by Griebler, Stein, et al. (2014). According to this scheme, monitoring wells with more than 70 % Crustaceans and less than 20 % Oligochaetes have a 'natural' status (indicating very good or good ecological conditions). Considering the entire study period, most monitoring wells in this study fall within this category (Figure 5c).

Abiotic parameters show mostly constant conditions in the individual wells (Figure 5d, e, f). One notable exception concerning electrical conductivity is Efringen, where electrical conductivity values decrease over time (Figure 5f). In terms of well water temperature (Figure 5d), several sites show an increasing trend, e.g. Weingarten, Riedlingen and Furtwangen. More pronounced temperature changes are present in Kadelburg with temperature variations of $> 5\,°C$ between individual measurements and Furtwangen with an increase of 4.6 K between 2002 and 2020.

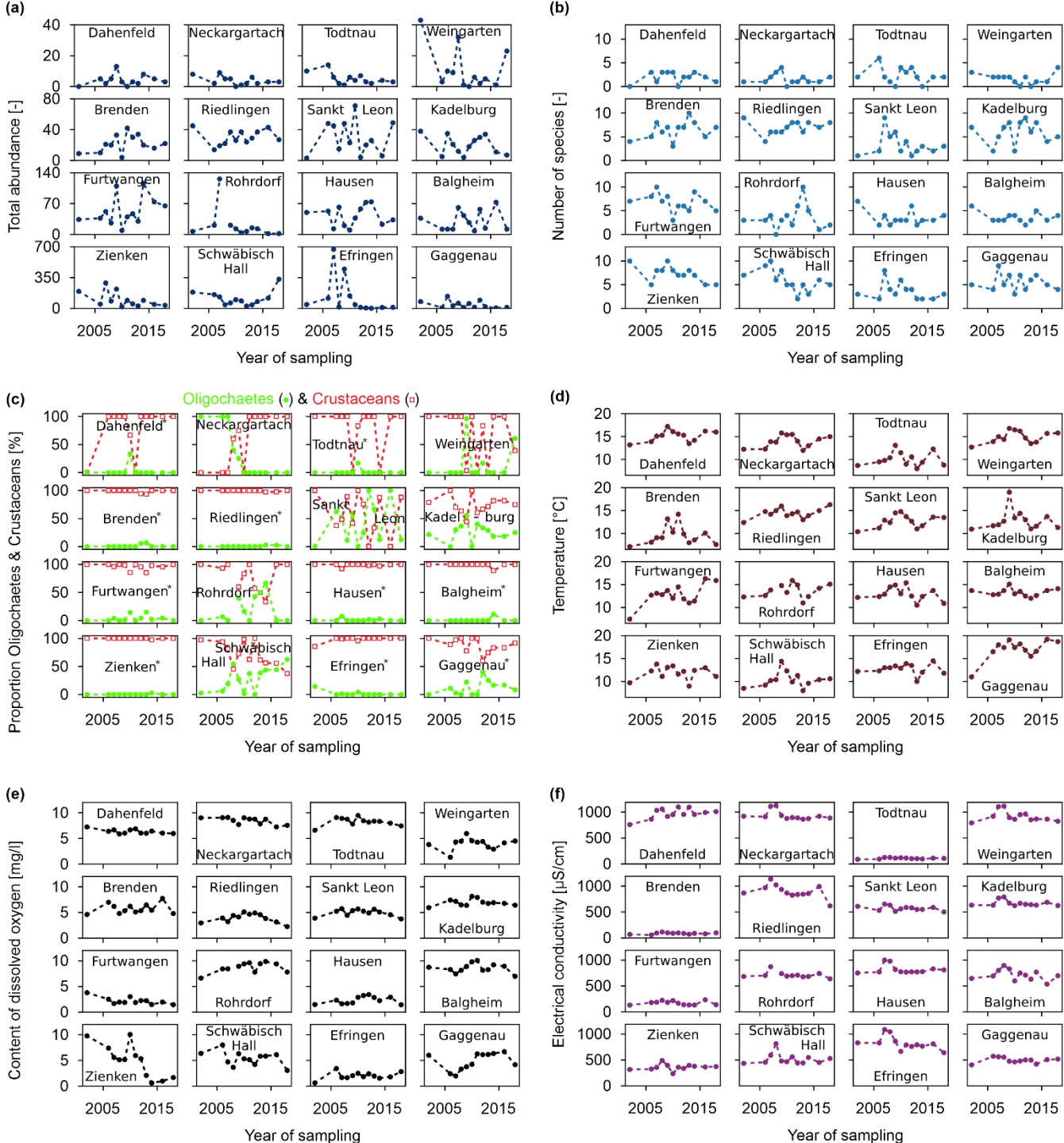

Figure 5: Time series of individual monitoring wells for (a) total abundance (please note the different scales of the y-axis); (b) number of species; (c) proportion of Crustaceans and Oligochaetes (* wells with a very good or good ecological status according to the

 **assessment scheme of** Griebler, Stein, et al. (2014))**; (d) temperature of the well water; (e) content of dissolved oxygen of the well water and (f) electrical conductivity of the well water.**

### 3.3  Statistical analysis

Correlation analyses are performed using the Spearman rank correlation coefficient to identify potential relationships between faunal and abiotic parameters, such as well configuration, geology and groundwater quality. In total, 164 parameters are considered, 92 of which are biotic parameters (65 taxonomic parameters and 27 other biotic parameters; see Table S4). Moreover, 72 abiotic parameters from the chemical-physical parameters of the LUBW monitoring are used. Samples are considered significantly correlated if the coefficient is $> 0.5$ or $< -0.5$.

The correlation coefficients reveal that the content of dissolved organic carbon (DOC) of the sampled aquifer water significantly correlates with the amount of biologically usable sediment ($\rho = 0.53$). Both parameters reflect the food content delivered through surface water input. Moreover, the DOC content correlates with the number of individuals and species of the subclass Ostracoda ($\rho = 0.51$; $\rho = 0.53$) and the number of juvenile Copepods ($\rho = 0.51$). Furthermore, there are correlations between the measured depth of the well and the abundance of the subclass Ostracoda and the total number of species ($\rho = 0.51$; $\rho = 0.51$). Also, some statistically significant correlations exist between faunal parameters and several anions and cations. These correlations are not further addressed here since only limited data is available for those parameters (mercury, strontium, etc.).

A PHATE-analysis is conducted to visualise the high-dimensional data set in low dimensions. As can be seen in the visualisation, three distinct groups can be identified (Figure 6). Group I contains monitoring wells in the karst aquifer of the Muschelkalk and Lettenkeuper formations in the northeast of BW and a fissured aquifer in the southern Black Forest, which generally have a low abundance. In contrast, Group II includes samples in all other wells, with the remaining wells of the southern Black Forest (Brenden, Furtwangen) located at the edge of this group. Group III consists of samples that had no groundwater fauna.

Temporal variations at the individual sites are also noticeable in the PHATE results by observing the spread of samples of individual locations. Samplings of very stable locations (e.g. Todtnau, Dahenfeld, Balgheim, and Riedlingen) with small parameter fluctuations over time are spatially more concentrated in the PHATE graph than wells with unstable conditions (e.g. Schwäbisch Hall, Weingarten). Generally, sites with stable conditions for both faunal and parameters (as identified in Table S3) are concentrated in the lower right area of Group II as Sub-Group II.a (except for Dahenfeld and Rohrdorf), and the upper-left area as Sub-Group II.b (Figure 6). The categorisation of measurement sites into stable and unstable according to their condition is based on the variance of faunistic and abiotic (hydrogeochemical) parameters over the period under investigation (Table S3). The variance is parameter-specific and depends on natural and seasonal fluctuations, such as groundwater temperature varying by 1 - 2 °C over the course of a year (Gibert et al., 1994; Taylor and Stefan, 2009b). Measurement wells with a total standard deviation over all observed parameters above 14 are classified as unstable wells.

While the affiliation of the wells with a specific group is constant over time (except for the mentioned samplings with no fauna), the location of different samplings within the group can be associated with concrete changes in specific parameters, specifically abundance, sediment content and temperature. Despite stable overall conditions, the well in Rohrdorf shows one outlier further down in the graph (orange triangle) representing the measurement in 2007 with significantly more individuals (126) than in other years. The amount of sediment is responsible for a clustering of samplings from different wells (Hausen, Gaggenau, Weingarten, among other) in the upper left corner of Group II, which are from 2009, 2011 and 2012 and lack information about the amount of sediment. The samplings Gaggenau (dark pink dots) show a further outlier in 2002, located more to the right, which is related to a relatively low sediment content < 1 ml. The same applies to Schwäbisch Hall with an outlier (red dots) to the right side of the graph, containing also a low sediment content, little detritus and low abundance (8 individuals). The samplings in Furtwangen (yellow triangles) are also spread over a wide area, with the ones at the bottom of Sub-Group II.a from 2002 and 2020 exhibiting significantly lower temperatures (4.3 °C and 8.7 °C), which are close to the measured temperatures in Brenden (light pink).

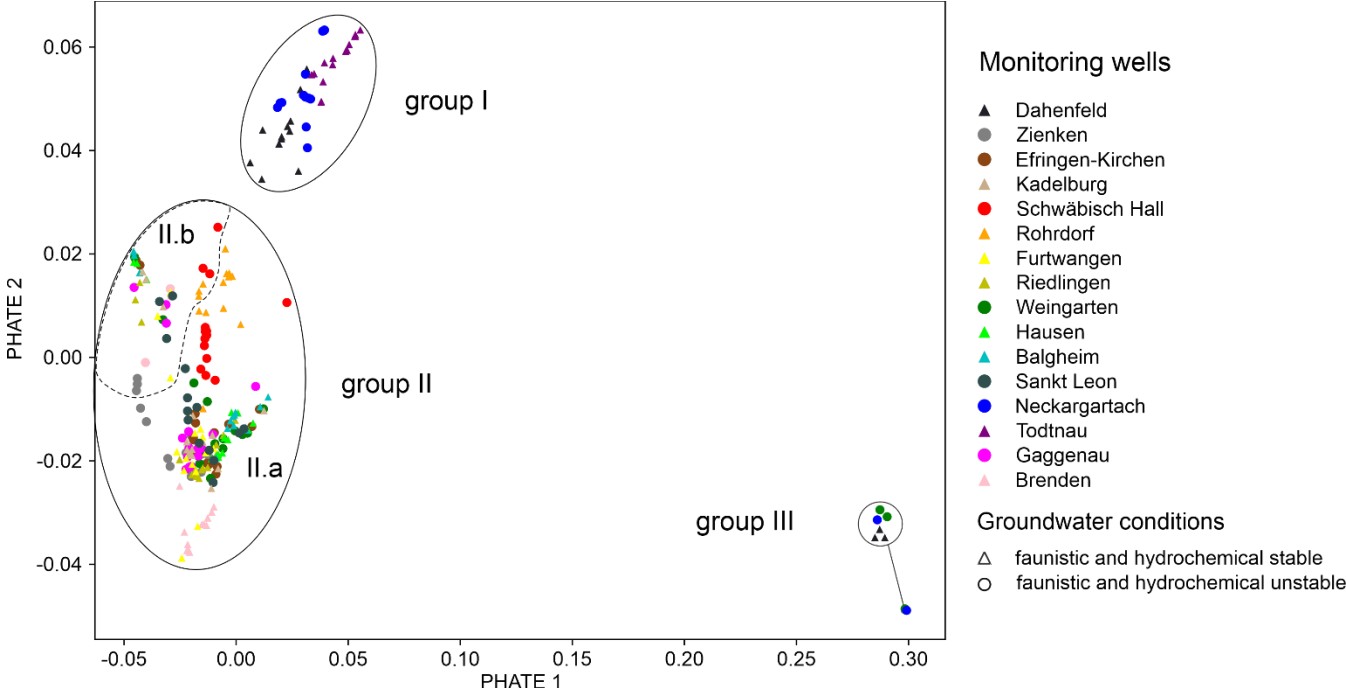

**Figure 6: Graphical result of the PHATE (Potential of Heat-diffusion for Affinity-based Trajectory Embedding)-analysis for all 16 monitoring wells from 2001/2002 to 2020, presented by the affiliation of the monitoring wells of each sampling.**

### 3.4 Local scale analysis

As described above, certain changes in faunal parameters in individual wells can be related to changes in abiotic parameters, for example, to a decreasing dissolved oxygen content as in Zienken, while other changes, such as the fluctuating ecological status in Neckargartach and Sankt Leon, cannot directly be related to varying abiotic parameters (Figure 5). Hence, we assess these two unstable wells in more detail with respect to changes at the surface surrounding of the wells and compare them to the well in Todtnau, which shows very stable conditions.

**Neckargartach**

The well in Neckargartach is 35 m deep with filter screens between 27 and 34 m in a fractured and karst aquifer formed by a clay-containing limestone (Figure 2). The overall number of individuals is very low (maximum of nine individuals per sampling) and decreases during the observation period (Figure 7). Faunal analysis on species level shows a distinct change in the faunal community and dominating species between 2002 and 2020. Between 2002 and 2007, Oligochaetes dominate the

faunal community (Figure 7). Most of these individuals belong to the species *Dorydrillus michaelseni*, which is only occasionally found in groundwater and is an indicator of slightly contaminated groundwater (Moog, 2002). In 2008 and 2009, the dominating species was *Chappuisius inopinus* (*Crustacea: Harpacticoida*), while from 2011 onwards, mainly Crustaceans were found, with the rare species *Parabathynella badenwuerttembergensis* being the dominating species. The temperature of the sampled well water fluctuates between 11.3 and 15.8 °C yet without a specific trend (see also Figure 5d). The nitrate

content decreases from 51.4 to 25.9 mg/l during the study period (Figure 7), and dissolved oxygen content decreases slightly from 10.9 to 8.0 mg/l (Figure 5 & Figure S2).

**Neckargartach**

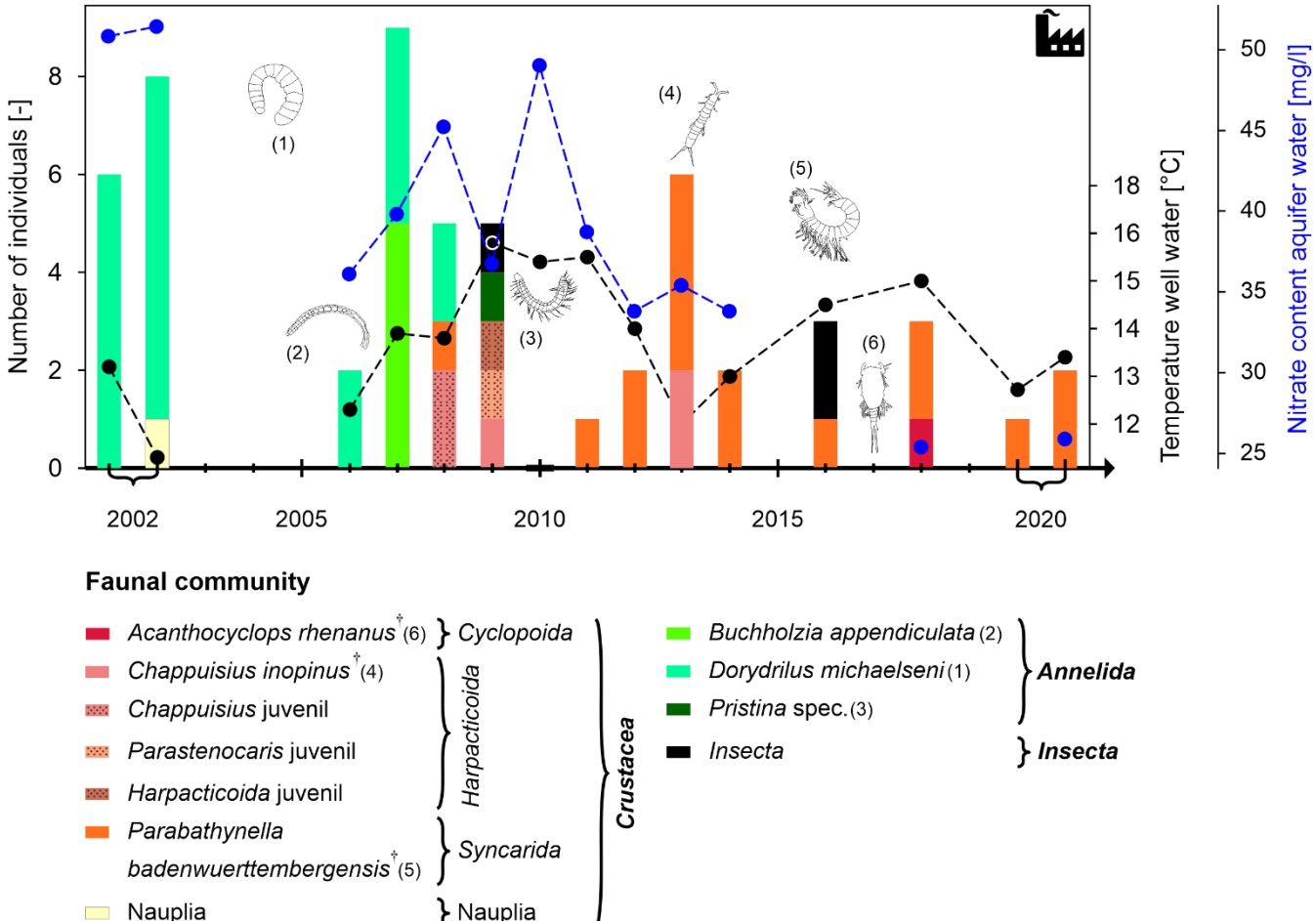

**Faunal community**

| | |
|---|---|
| ▪ (dark red) *Acanthocyclops rhenanus*[†](6) } *Cyclopoida* | ▪ (light green) *Buchholzia appendiculata* (2) |
| ▪ (salmon) *Chappuisius inopinus*[†](4) | ▪ (teal) *Dorydrilus michaelseni* (1) |
| ▪ *Chappuisius* juvenil | ▪ (dark green) *Pristina* spec. (3) |
| ▪ *Parastenocaris* juvenil | ▪ (black) *Insecta* |
| ▪ *Harpacticoida* juvenil | |
| ▪ (orange) *Parabathynella badenwuerttembergensis*[†](5) } *Syncarida* | |
| ▪ (pale yellow) Nauplia } Nauplia | |

*Harpacticoida* · *Crustacea* · *Annelida* · *Insecta*

**Figure 7: Temporal development of the faunal community (abundance and composition of the faunal community; higher taxa in bold letters; [†] stygobiont species) and the well water temperature (secondary y-axis) at the bottom of the monitoring well in Neckargartach during the period of investigation (2002 – 2020). No sampling was conducted in years with no bar.**

**Sankt Leon**

The well Sankt Leon is located next to a field path between a forest and an agricultural area near the village of Sankt Leon (Figure S3a), with filter screens between 4 and the well bottom in 10 m, in the quaternary glacial sand and gravels of the Upper Rhine Valley (Figure 2). The well shows unstable abiotic and faunal conditions (between 0 and > 400 individuals per sampling, Figure 5) and significant variations in the faunal compositions (Figure 8).

Common taxa in this well are Nematodes, as well as different Cyclopoids (*Crustacea*). In the first decade, rather ubiquitous species colonise the well in large numbers, with *Graeteriella unisetigera* and *Diacyclops languidoides* (*Crustacea*: *Cyclopoida*) being the dominating species. The latter is one of Germany's most common and widespread groundwater species (Matzke et al., 2009; Schminke et al., 2007). From 2010 onwards, fewer individuals and different taxa can be found, with

*Bathynella freiburgensis* (*Crustacea*: *Syncarida*) being the dominating species in recent years. Well water temperature shows significant fluctuations between 11 °C and 14 °C, yet without visible trend. The nitrate content decreased from 70 mg/l to below 20 mg/l (Figure 8).

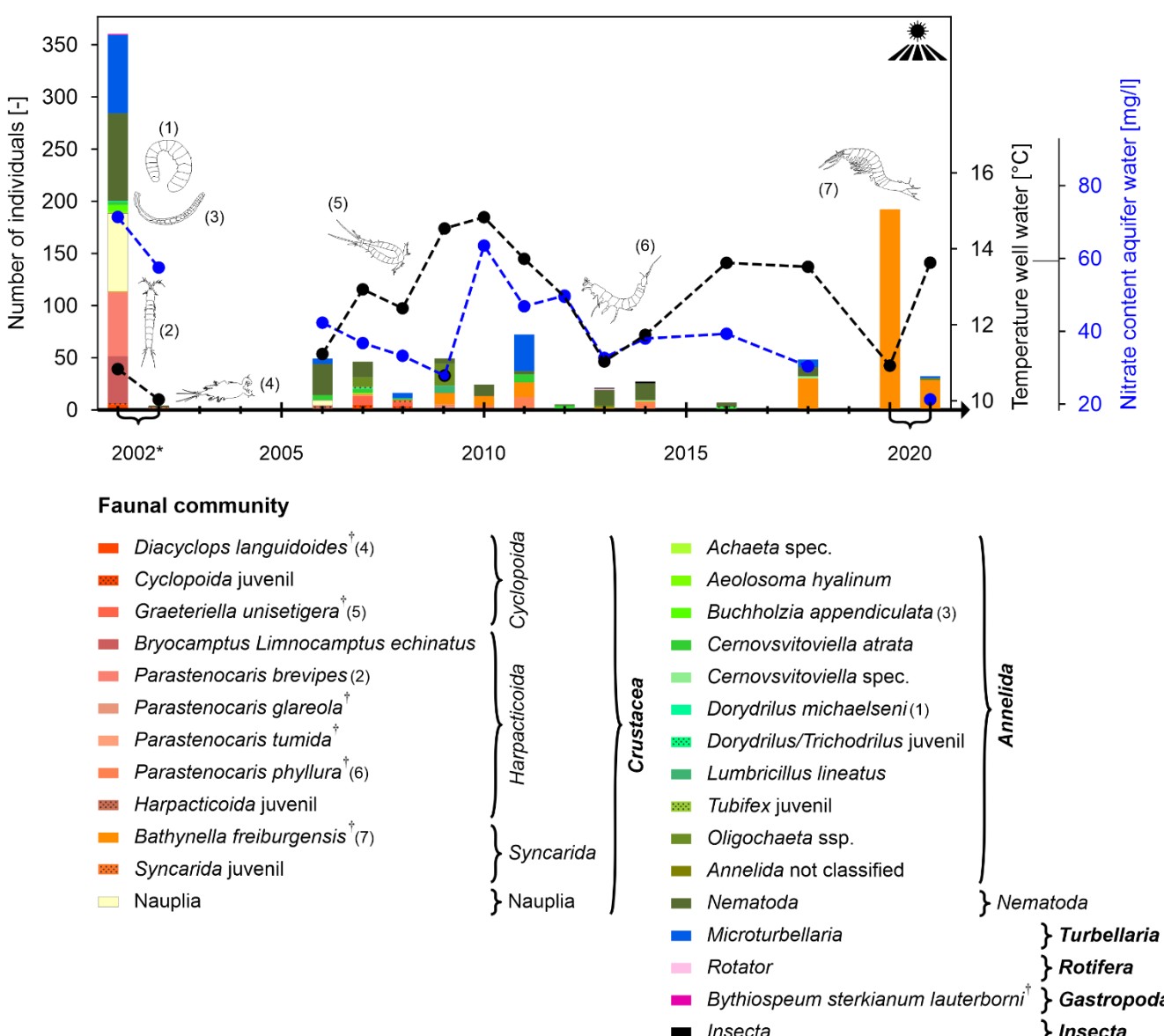

**Figure 8: Temporal development of the faunal community (abundance and composition of the faunal community; higher taxa in bold letters; † stygobiont species), of the content of nitrate (data of the LUBW, blue y-axis on the right) and the well water temperature (black y-axis on the right) on the bottom of the well in Sankt Leon between 2002 (* fauna sampling in May and December) and 2020. No sampling was conducted in years with no bar.**

**Todtnau**

The well in Todtnau is also located in an agricultural area near a small stream (Prägbach) and the village of Todtnau-Geschwend (Figure S3b), with a depth of 39 m and well screens between 5 and 36 m in a fractured crystalline aquifer in the southern Black Forest. The number of individuals and species is generally low (Figure 9). The well in Todtnau is colonised by stygobiont species only, except for *Collembola* (*Insecta*) in two years, which live in the soil and on the water's surface. The most common species during the past 20 years is *Crangonyx subterraneus* (*Crustacea*: *Amphipoda*), which is widespread,

ecologically very flexible and inhabits all kinds of underground habitats, but prefers fractures and gravels (Schminke et al., 2007). The species *Proasellus cavaticus* (*Crustacea*: *Isopoda*) is also commonly present at this location. Individuals of this species can be up to 1 cm long and, therefore, prefer larger cavities, e.g. caves and fractured aquifers, such as the one in Todtnau. Another common species is *Niphargus auerbachi* (*Crustacea*: *Amphipoda*), a comparably big, stygobiont groundwater species. The groundwater species *Troglochaetus beranecki* (*Annelida*: *Polychaeta*) found in 2001, 2007 and 2011

is cold-stenotherm, has low water chemistry requirements (Schminke et al., 2007) and also inhabits deep groundwater wells (Matzke et al., 2009). It is also noted that juvenile crustaceans, especially Niphargids, are predominant in the last two years of sampling. Well water temperatures are generally low due to higher altitude in the Black Forest. The same applies to electrical conductivity related to filter screens in crystalline rocks and organic matter content due to the depth of the well.

**Todtnau**

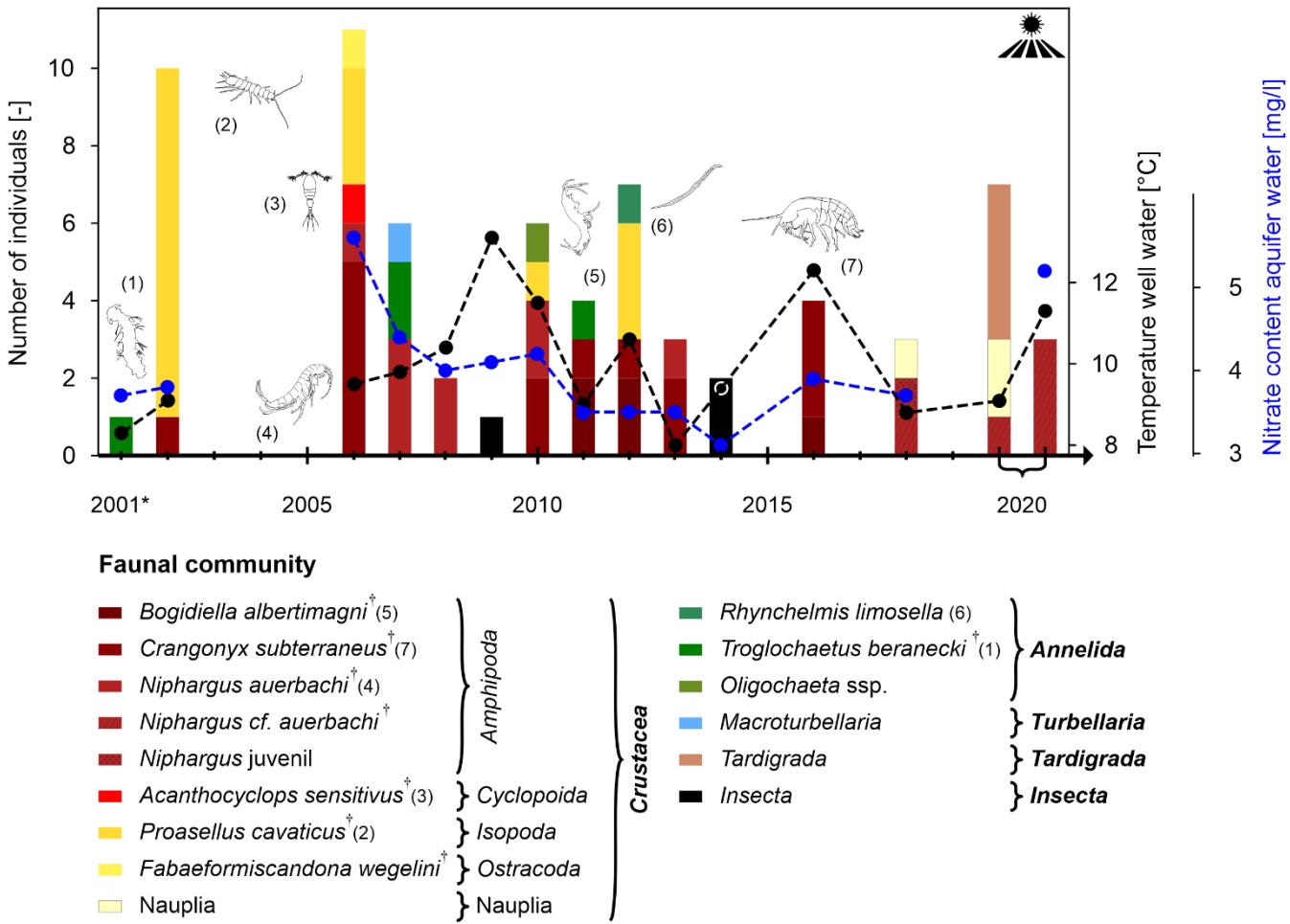

Figure 9: Temporal development of the faunal community (abundance and composition of the faunal community; higher taxa in bold letters; † stygobiont species), of the content of nitrate (data of the LUBW, blue y-axis on the right) and the well water temperature (black y-axis on the right) on the bottom of the well in Todtnau-Geschwend between 2001 (* fauna sampling in November) and 2020. No sampling was conducted in years with no bar.

## 4. Discussion

### 4.1 Faunistic overview

Overall, there is no regional pattern in the spatial distribution of faunal abundance and biodiversity (Figure **3**). However, there is a visual relationship between abundance and biodiversity, as monitoring wells with many individuals also show more species (and vice versa). Potential reasons for this are a high hydrogeological and hydro-chemical heterogeneity in combination with a small number of monitoring wells, the time between sampling, superimposing effects of local influences and site-specific parameters linked to variations in topography and geology. The spatial distribution of the faunal community aligns with

findings of distribution patterns of groundwater fauna from previous studies (Fuchs, 2007; Fuchs et al., 2006; Hahn and Fuchs, 2009; Stein et al., 2012, 2015). According to Fuchs et al. (2006), for example, the main distribution focus of Syncarids is also in the north of Upper Rhine Valley and the catchment of the Neckar River.

## 4.2 Temporal analysis

There is no observable temporal change in the average total abundance or its variation, in the number of species, which indicates that the large-scale biodiversity was stable over the past two decades (Figure 4). Moreover, no significant changes are revealed over time for the abiotic parameters, suggesting stable hydro-chemical conditions over time. Outliers in individual years, such as in 2007, hint at a more pronounced surface influence and disturbed conditions, particularly in Efringen, Rohrdorf, Zienken, Schwäbisch Hall, and Gaggenau.

While there is no apparent trend on the regional scale, there are significant variations between individual years for some individual wells, indicating more complex temporal behaviours on local scale (Figure 5). Gradual changes in the faunal community from stygobiont species to multiple stygobiont and stygophile species are potentially linked to surface water influence (Schminke et al., 2007). For example, in Schwäbisch Hall, where the well appears to be intact and no anomalies were found during sampling, it is more likely that changes in the faunal community and abundance are related to the dried-out

lakes in the wooded area, but potential contaminations cannot be ruled out. Decreasing dissolved oxygen contents, as in Zienken, could be related to microbial oxygen degradation due to a high organic matter input. Furthermore, a strong surface influence in Zienken is linked to periods with higher numbers of ubiquitous species from the surface and a higher bacterial count, as also observed by Stein et al. (2015). Fluctuations of abundance and number of species in several wells indicate unstable conditions likely caused by varying surface influence. Thus, species appearing only sporadically at one location can

indicate an intermittent surface water input into this well, as can be seen in Gaggenau in the increase in oxygen concentration after 2006 and the higher number of Oligochaetes and Nematodes. These results are consistent with previous studies, which showed that groundwater communities strongly depend on the hydrologic exchange with surface water (Fuchs et al., 2006; Gutjahr et al., 2014; Hose et al., 2015).

The ratio of Crustaceans and Oligochaetes is an important parameter for the ecosystem status. Only six wells have a proportion

of Oligochaetes > 20 % in one or multiple years, indicating disturbed conditions: Kadelburg, Schwäbisch Hall, Rohrdorf, Sankt Leon, Weingarten and Neckargartach (Figure 5c).

While the wells in Kadelburg, Schwäbisch Hall, Rohrdorf and Sankt Leon show a fluctuating ecological status over time, probably related to varying surface water influence, the well in Neckargartach shows a distinct change in the ecological status in 2008, which does not seem to be related to other biotic or abiotic parameters. Abiotic parameters show mostly constant

conditions in the individual wells. The increase in electrical conductivity in Efringen is most likely caused by the reconstruction of the well from an underfloor to a surface observation well (see Figure 5c).

Only in terms of well water temperature, several sites show an increasing trend, which is in the same range as observed changes in groundwater temperature in previous studies (Figura et al., 2011b; Menberg et al., 2014b). More pronounced temperature changes are more likely related to varying surface influence or environmental conditions during measurements.

Summing up, faunistic and abiotic parameters are mostly constant over time in the individual wells of the study area. Nine out of 16 wells (Dahenfeld, Balgheim, Hausen, Riedlingen, Brenden, Kadelburg, Rohrdorf, Furtwangen, Todtnau) show stable conditions over time concerning the variance of the faunistic and abiotic parameters (Table S3). In contrast, seven wells (Weingarten, Schwäbisch Hall, Efringen-Kirchen, Zienken, Gaggenau, Neckargartach, Sankt Leon) show higher standard deviations and, therefore, unstable conditions. These are often linked to a varying influence of surface water, which aligns with

previous studies (Dole-Olivier, 1998; Foulquier et al., 2011; Stein et al., 2012).

## 4.3 Statistical analysis

The observed correlation between the DOC and Ostracodes can be explained by the fact that the class Ostracoda rely on a temporary surface influence, as they need nutrients (mainly detritus) from the surface and are thus typical for the interstitial and porous aquifers (Mösslacher and Hahn, 2003). A positive relationship between depth and the number of Ostracodes is also

observed in Reeves et al. (2007). Moreover, these findings are consistent with a positive relationship between invertebrate density and dissolved and particulate organic matter found in previous studies (Datry et al., 2005; Hahn, 2006; Mösslacher and Notenboom, 1999). Besides the DOC, no correlations are found between faunal parameters and those reflecting food supply for groundwater ecosystems. Thus, food is not a limiting factor for the occurrence of groundwater fauna in the studied wells. In general, only a few significant correlations were found. This is in line with observations by previous studies where

correlation analysis revealed significant correlations between chemical variables but no or very few and only weak correlation between abiotic and biotic parameters (Dumas et al., 2001; Griebler et al., 2014b; Hahn, 2006; Koch et al., 2021; Schmidt and Hahn, 2012; Steube et al., 2009). A commonly noted reason for this is that ecosystems are complex multivariate systems exposed to multiple stressors simultaneously (Steube et al., 2009).

In the PHATE analysis, the affiliation of the wells with a specific group is generally constant over time and the clustering of

wells with stable conditions confirms the categorisation of the wells using the variance of their faunistic and hydro-chemical parameters. In addition, the PHATE analysis shows that biodiversity, illustrated by the number of taxa and individuals, and geological conditions, such as the type of aquifer, have the largest impact on the clustering of the monitoring wells into three distinct groups. Although, the correlation analysis does not indicate any relationship between the amount of sediment and other parameters, except the proportion of biologically usable sediment, a trend in sediment content in the PHATE-analysis is visible.

This trend is due to very local effects over time in conjunction with other fluctuating parameters.

Previous studies also showed that hydrogeological parameters strongly influence the occurrence and composition of groundwater fauna (Koch et al. 2021; Stein et al. 2012). Geology and hydrological connectivity significantly influence water chemistry and habitat availability and, therefore, biotic distribution (Hahn 2006, Fuchs 2007; Fuchs et al. 2006; Korbel, Chariton, and Hose 2018; Korbel and Hose 2015; Tione, Bedano, and Blarasin 2016). For instance, the abundance and species

richness of crustacean fauna in the alluvial aquifers are most related to hydrological conditions, oxygen concentrations and geologic structures (Mösslacher, 1998), which is consistent with our findings. Moreover, this is consistent with the results from Korbel et al. (2018), who state that 'sediment size, and thus the size of interstitial voids, is a key limiting factor' for stygofauna.

These findings show that the amount of sediment can be used as an indicator for the pore structure, which determines hydraulic conductivity and living space (Mösslacher, 1998), and can thus be used as a proxy for living conditions. Temperature, on the other hand, can be used as an indicator for surface influence, as it is an indirect marker of the degree of hydrological exchange with the surface (Schönborn 2003, Hahn 2006). This was also observed by Koch et al. (2021) on the city scale, where local geology influenced the occurrence of groundwater fauna, the number of individuals and the food supply. Other studies also observed a significant effect of groundwater temperature on fauna, as diversity decreased with increasing temperatures in laboratory and small-scale field studies (see Brielmann et al. 2009; Spengler 2017).

### 4.4 Local scale analysis

**Neckargartach**

Overall, the observed faunal changes in Neckargartach represent a change from stygophil species to domination of stygobiont species, as well as a simultaneous change from dominating Oligochaetes to Crustaceans and thus to the aforementioned change in the ecological status (Figure 5c). The observed faunal changes also coincide with alterations in land use of the surrounding area (Figure 10). After the first period with dominating Oligochaetes, a previously unsealed area was first converted into a gravel-covered car park in 2008 and later into an industrial warehouse in 2018. During that time, different Crustaceans species began to dominate the faunal community (Figure 10). Surprisingly, this surface change is not reflected in the well water temperatures, even though groundwater under covered surfaces typically shows higher temperatures than under unsealed surfaces (Tissen et al. 2019 ERL). In Neckargartach, this is likely linked to groundwater inflow from the adjacent agricultural area into the monitoring well (Lang et al., 2004). The nitrate content decreases most likely due to the absence or reduction of fertilisation. Also, observed changes in nitrate and oxygen content are consistent with the shielding from the surface as main source of oxygen and the change from stygophil to stygobiont species (Bork et al., 2009; Hahn, 2006; Malard et al., 1996; Mösslacher, 1998; Pospisil, 1999; Sket, 1999).

Thus, there is a clear link in Neckargartach between decreasing surface influence, caused by increasing surface sealing, and dissolved oxygen content as well as the composition of groundwater ecosystems, as also observed by Korbel et al. (2013). In this study, agriculture in areas with different land use affects groundwater ecosystems (composition of stygofauna and microbial assemblages) due to changes in groundwater quality (nitrate and phosphorus contents). Hence, a higher abundance of Cyclopoids, Harpacticoids and Oligochaetes was found under irrigated areas. However, it has to be mentioned that the improvement of the ecological status according to Griebler, Stein, et al. (2014) through surface sealing is a site-specific

observation for Neckargartach. Generally, surface sealing and the related decrease in dissolved oxygen will more likely lead to the deterioration of groundwater ecosystems (Hervant and Malard, 1999; Korbel et al., 2022; Mösslacher, 1998).

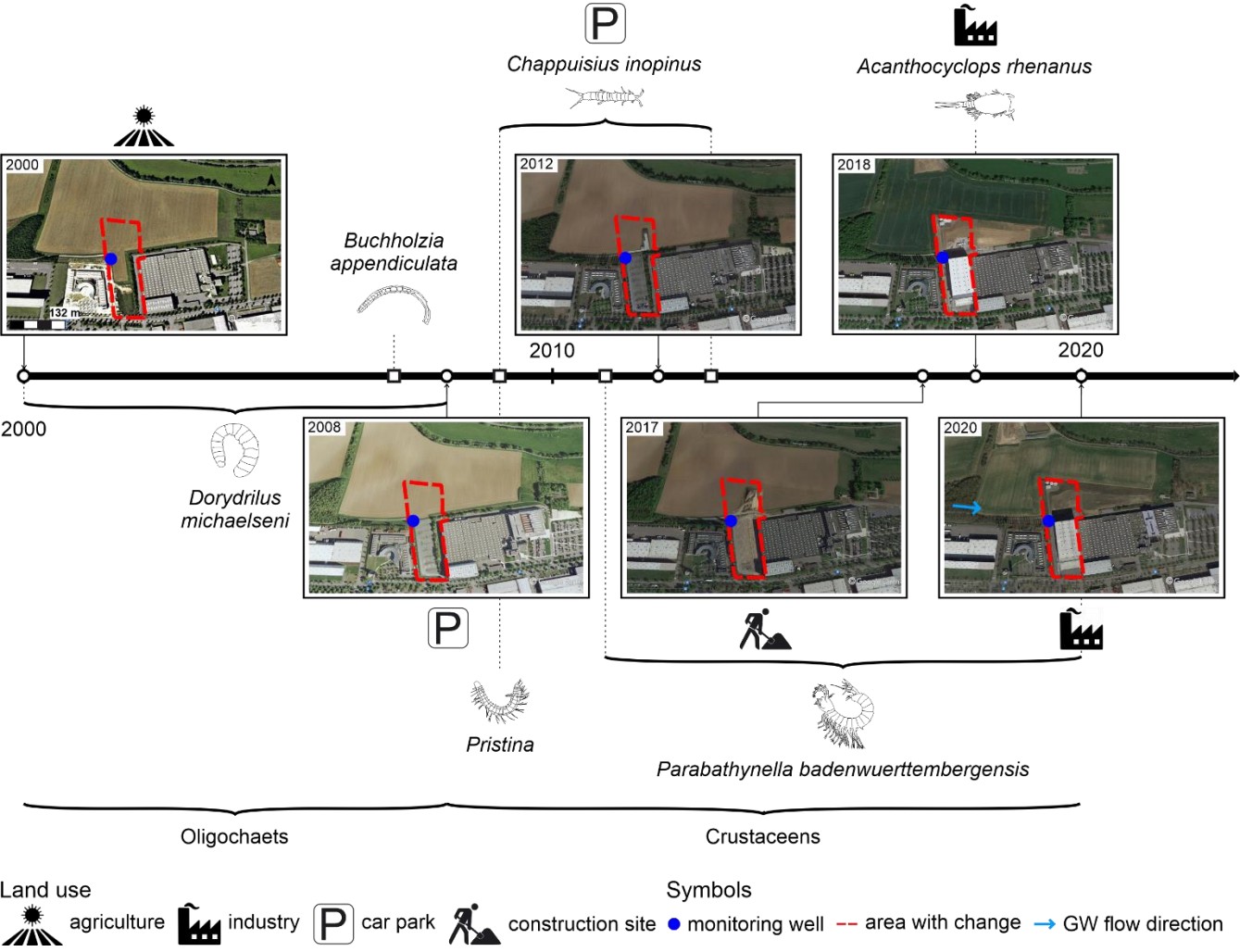

**Figure 10: Temporal changes of land use using aerial image interpretation and faunal community structure of the monitoring well**
**Neckargartach in the industrial park Böllinger Höfe between 2000 and 2020 (image source: Google Earth Pro** (Google LLC., 2022)**).**

**Sankt Leon**

Although these faunal changes indicate a weakening surface influence over time, no changes in land use or surface conditions were observed (Figure S3). As *Bathynella freiburgensis* tolerates a large range of temperatures and is typical for the Upper Rhine Valley (Fuchs, 2007; Spengler, 2017), individuals of that species might have a competitive advantage over other species. High nitrate contents in the first years are most likely linked to intensive agricultural fertilisation, in particular for asparagus cultivation, which is typical for this region. However, these fluctuations do not seem to correlate with the observed faunal

shifts. Furthermore, previous studies showed that a nitrate concentration below 50 mg/l has no direct impact on groundwater fauna (Fakher el Abiari et al., 1998; Di Lorenzo et al., 2020; Di Lorenzo and Galassi, 2013; Mösslacher and Notenboom, 1999). Accordingly, more detailed and site-specific investigations (e.g. more accurate, in-situ groundwater temperature and dissolved oxygen measurements over depth) would be needed to clarify the drivers of the unstable conditions in Sankt Leon.

**Todtnau**

In contrast to the two wells discussed previously, the well in Todtnau shows stable conditions (Table S3, Figure 5). Despite its location in an agricultural area, the nitrate content is below the geogenic background (Figure 9). However, despite stable physico-chemical conditions and no change in land use or surface conditions, certain changes in the faunal community can be observed as different species occur in different years, yet without any visible trends. These are also the samplings that show a non-natural status of the ecosystem according to Griebler, Stein, et al. (2014) due to the absence of Crustacea (Figure 5c). The predominance of juvenile crustaceans, especially Niphargids, may indicate a recovery of the population after a lack of colonisation in 2014. Years without colonisation may point to bad living conditions with a lack of nutrients and, therefore, challenging living conditions. In general, the faunal composition in Todtnau reflects the hydrogeological conditions of the well, with many species typical of large cavities and higher altitudes.

## 5. Conclusion

This study analyses the faunal composition and abiotic parameters of 16 groundwater measurement wells in the German state of Baden-Württemberg over the past two decades. Statistical analyses are used to distinguish wells with stable and unstable faunal conditions and to assess the impact of specific hydrological and hydro-chemical parameters on this characterisation. Time series of individual wells are also discussed in combination with past aerial images to analyse the impact of changes in surface conditions.

Considering the entire study area, we observed no long-term changes or trends in abiotic or faunal parameters, indicating generally stable and ecological good conditions. However, temporal fluctuations in faunal parameters, such as total abundance and number of species, and thus unstable conditions are observed for seven out of 16 wells. In some cases, these changes are directly related to gradual changes in abiotic parameters, such as decreasing abundances due to reduced dissolved oxygen contents. Yet more often, there are no clear patterns in individual abiotic parameters; instead, superimposing effects of multiple parameters linked to increasing or weakening surface influence lead to changes in groundwater fauna. Results from a multivariate PHATE analysis confirm these findings and highlight the hydrogeological setting, the content of sediment and detritus in the well and the temperature as influential factors.

Examining faunal changes on species level for selected wells reveals that unstable conditions can be linked to changes in surface sealing by anthropogenic construction measures, which even changed the ecological status at one specific site. However, variable faunal composition and fluctuating abundances were also observed for sites with no visual changes in land

use and surface influence and also (although less prominent) for a deep, well-shielded site with very stable abiotic groundwater conditions. Thus, more long-term studies of groundwater ecology with higher spatial and temporal resolution are necessary to
further improve our understanding of faunal shifts over time.

These findings have direct implications for large-scale biomonitoring in groundwater, which is becoming increasingly important. Transferability of local observations to a larger scale is very limited due to small-scale heterogeneities in hydrogeological conditions and superimposing, site-specific effects. Noticeable environmental changes for wells in the state of Baden-Württemberg were often linked to changes in dissolved oxygen content, well water temperature and sediment
content. Accordingly, these parameters should be accurately and representatively measured in the water column of the well (if possible depth-resolved) and assessed in combination with hydrogeological and surface conditions to obtain more reliable, representative and robust biomonitoring results. Limitations regarding the sampling method must be also considered when interpreting faunal results. This will also help to design management tools for agencies and local authorities.

As it was shown, various parameters, such as surface conditions (built-up areas, underground infrastructure, sealing, etc.),
hydrogeological (type of aquifer, size of pore cavities, groundwater flow, etc.), and physico-chemical (temperature, content of dissolved oxygen, nutrient supply, etc.) influence the health of the ecosystem. Thus, future assessment schemes should consider a more comprehensive range of indicators. Moreover, future groundwater fauna sampling campaigns should employ shorter sampling intervals, e.g. on a monthly basis, to also address effects of seasonality. Furthermore, the observed faunal fluctuations in wells in natural, unaffected areas with stable abiotic conditions stress that reference locations for ecological groundwater
assessments and biomonitoring have to be carefully selected, ideally on multi-annual data. Therefore, reference sites should comprise different settings, including forests, green areas, cities, industrial areas, and surface waters, to account for small-scale heterogeneities in hydrogeological conditions and land use. This is crucial for the transferability of findings to larger scales and longer time frames, as well as to identify sites at high risk.

*Data availability.* Results of the physical–chemical, faunistic and statistical analysis are available at DOI: 10.35097/m4gepnhuc3yengju.

*Supplement.* The supplement related to this article is available online at:

*Author contributions.* PB, KM and HJH provided the topic and supervised the work. AF and FK executed the fieldwork and AF evaluated the samples. FK evaluated the collected data, interpreted and visualised the results and wrote the first draft of the paper. KM, AF, HS, HJH and PB edited the paper.

*Competing interests.* The authors declare that they have no conflict of interest.

*Acknowledgements*. We would like to thank the Regional Office of Environment Baden-Württemberg (LUBW), especially in the person of Klaus-Peter Barufke, for their support. Moreover, we would like to thank Cornelia Spengler (IGÖ GmbH) for her support at the beginning of the study. We acknowledge support from the KIT Publication Fund of the Karlsruhe Institute of Technology.


*Financial support*. Funding for the present work was provided by the State-Graduate-Scholarship (Landes-Graduierten-Förderung LGF) (Fabien Koch), the Margarete von Wrangell-program of the Ministry for Science and Art (MWK) Baden-Württemberg (Kathrin Menberg) and the German Federal Environment Foundation (DBU, AZ 3392) in the framework of the project 'Thermostress'.

The article processing charges for this open-access publication were covered by the Karlsruhe Institute of Technology (KIT).

*Review statement*. This paper was edited by Jan Seibert and reviewed by two anonymous referees.

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
