# Peer review of "Temporal shift of groundwater fauna in South-West Germany"

_Hydrology and Earth System Sciences, 2024_

## Author Response (AR1)

Dear Reviewers, dear Editor,

we would like to thank you for your time and the constructive comments, which helped to improve the quality of the manuscript. Please find our detailed replies on the comments below. We hope that we have answered all your remarks.

In general, our replies to the referee's comments are highlighted in blue. To highlight the nature of our replies we use a traffic light system indicating agreement with the referee marked in green, partial agreement in yellow, and objections in red.

**Reviewer #1**

**Major Comments**

1) The MS provides a thorough overview of the study, including the use of long-term groundwater data to identify shifts in groundwater fauna due to natural or anthropogenic impacts. However, providing specific examples or anecdotes from the research could help to make the abstract more engaging and informative.

   Response: We agree and added more details on the findings and site-specific content:

   "By examining aerial images of the surroundings of individual wells, we found that anthropogenic impacts, such as construction sites and surface sealing, can cause significant shifts in groundwater fauna and changes in the ecological status in positive as well as negative directions. However, variable faunal composition and abundances were also observed for sites with very stable abiotic conditions in anthropogenically less affected areas such as the Black Forest."

2) The MS could benefit from a clearer statement of the research objectives and hypotheses, to help readers better understand the purpose of the study.

   Response: We agree and rewrote the last paragraph of the introduction, which now reads:

"The main objective of this study is to identify changes in groundwater fauna in the German state of Baden-Württemberg over the last decades on both regional and local scales. We hypothesize that these changes can be related to natural or anthropogenic stress observable through changes in abiotic parameters, such as temperature increase from climate change and high nitrate concentration from agricultural fertilisation, as well as land use changes, such as an increase in surface sealing. To this end, multiple abiotic and biotic groundwater parameters of all wells are first jointly analysed over time, before individual sites are scrutinised for small-scale changes. Furthermore, changes in groundwater ecosystems on different spatial scales and the implications of the observed changes for biomonitoring are assessed. Finally, we identify ecological and physico-chemical parameters most suitable for robust biomonitoring."

3) The mention of the PHATE analysis could be expanded upon to provide more detail on this methodology and how it was applied in the study.

Response: We agree to add more information on the PHATE-analysis and how it is applied in this study. Thus, we added the following sentences respectively:

"In contrast to most existing methods, a PHATE-analysis allows to obtain biologically interesting structures from increasingly large, modern datasets while accounting for the high level of noise inherent in biological datasets."

"Another advantage of this method is that biological and abiotic parameters can be considered together, allowing identification of further dependency structures."

"The selection of parameters is based on previous studies, as well as empirical values on influencing factors from the literature (Hahn, 2006; Koch et al., 2021; Korbel et al., 2018; Stein et al., 2012)."

"From each well, data of all annual measurement campaigns are analysed separately to assess temporal variations in faunal communities and abiotic conditions."

4) The abstract mentions the importance of considering hydro(geo)logical changes and surface conditions in assessing changes in groundwater fauna, but it could be helpful to provide more

specific recommendations for future research or management strategies based on the study findings.

Response: We agree that it can be helpful to provide more specific recommendations for future research and management. Thus, we rewrote part of the conclusions as follows:

"Accordingly, these parameters should be accurately and representatively measured in the water column of the well (if possible depth-resolved) and assessed in combination with hydrogeological and surface conditions to obtain more reliable, representative and robust biomonitoring results. This will also help to design management tools for agencies and local authorities.

As it was shown, various parameters, such as surface conditions (built-up areas, underground infrastructure, sealing, etc.), hydrogeological (type of aquifer, size of pore cavities, groundwater flow, etc.), and physico-chemical (temperature, content of dissolved oxygen, nutrient supply, etc.) influence the health of the ecosystem. Thus, future assessment schemes should consider a more comprehensive range of indicators. Moreover, future groundwater fauna sampling campaigns should employ shorter sampling intervals, e.g. on a monthly basis, to also address effects of seasonality. Furthermore, the observed faunal fluctuations in wells in natural, unaffected areas with stable abiotic conditions stress that reference locations for ecological groundwater assessments and biomonitoring have to be carefully selected, ideally on multi-annual data. Therefore, reference sites should comprise different settings, including forests, green areas, cities, industrial areas, and surface waters, to account for small-scale heterogeneities in hydrogeological conditions and land use. This is crucial for the transferability of findings to larger scales and longer time frames, as well as to identify sites at high risk."

5) Clarifying the specific types of anthropogenic impacts that were observed, such as pollution sources or land use changes, could help to provide a more nuanced understanding of the study results.

Response: We agree and added more specific statements on anthropogenic impacts at different places in the manuscript:

Abstract: "… we found that anthropogenic impacts, such as construction sites and surface sealing, ...".

Introduction: "We hypothesize that these changes can be related to natural or anthropogenic stress observable through changes in abiotic parameters, such as temperature increase from climate change and high nitrate concentration from agricultural fertilisation, as well as land use changes, such as an increase in surface sealing."

Discussion: "High nitrate contents in the first years are most likely linked to intensive agricultural fertilisation, in particular for asparagus cultivation, which is typical for this region."

Conclusions: "As it was shown, various parameters, such as surface conditions (built-up areas, underground infrastructure, sealing, etc.), hydrogeological (type of aquifer, size of pore cavities, groundwater flow, etc.), and physico-chemical (temperature, content of dissolved oxygen, nutrient supply, etc.) influence the health of the ecosystem."

**Reviewer #2**

This is a valuable piece of research and, with some additional statistical analysis and clearer aims, the extensive dataset summarising changes in groundwater ecosystems over a long -period of time will contribute to the knowledge of groundwater ecosystem. I find the manuscript in its current form hard to read and in some points confusing, due to the use of English and the lack of structure. The three major concerns I have as summarised as 1. sampling methodologies & details  2. some statements are oversimplifying these complex ecosystems,  3. some correlations and inferences are being made without sufficient statistical evidence.

**Major Comments**

It is suggest that:

1)  The manuscript requires a thorough edit for English. There are too many issues to correct by a reviewer. There are numerous grammatical errors and use of words that make little sense in the context of the text.

Response: Thank you for pointing out grammatical errors and word usage. We agree and will check the manuscript again in this regard.

2)  Aims and research questions need to be more clearly articulated

Response: We agree and rewrote the introduction accordingly. (See Major Comment#2, Reviewer1)

"The main objective of this study is to identify changes in groundwater fauna in the German state of Baden-Württemberg over the last decades on both regional and local scales. We hypothesize that these changes can be related to natural or anthropogenic stress observable through changes in abiotic parameters, such as temperature increase from climate change and high nitrate concentration from agricultural fertilisation, as well as land use changes, such as an increase in surface sealing.  To this end, multiple abiotic and biotic groundwater parameters of all wells are first jointly analysed over time, before

individual sites are scrutinised for small-scale changes. Furthermore, changes in groundwater ecosystems on different spatial scales and the implications of the observed changes for biomonitoring are assessed. Finally, we identify ecological and physico-chemical parameters most suitable for robust biomonitoring."

3) The report structure makes the research hard to understand and the reader would benefit from a clearer structure addressing the studies aims (potentially in subsections). Concepts are scattered throughout the paper, for example temperature and sediment are discussed in line 233- however I cannot find these result anywhere until the discussion (4.4)- and there doesn't appear to be any statistical analysis of temperature and biota? There are some results presented in the results section, and others in the discussion section. The paper would benefit from a clear discussion section which is entirely separated from the results. This would allow the main aims of the paper to be discussed and compared to previous literature in one section, separated by clear sub-headings. For example, most of discussion section 4.4 is results not discussion, this should be moved to section 3.1 with results explained clearly in the discussion section following.

Response: We agree that a clear structure is helpful for the reader.

Results for temperature and sediment content measurements are indeed described and discussed accordingly in section 4.2 in terms of overall temporal development. In section 4.3, the interrelationship of temperatures and sediment with other abiotic and faunal parameters for all measurement wells is discussed. Finally, temperature effects are discussed for individual wells in section 4.4 (sediment content was not found to be relevant on this scale). In accordance with the rephrased aim of the study, the steps shown in the workflow (Figure 1) are now identical to the subheadings in the results, as well as in the discussion section in order to impose a clear structure, as well as clear separation between results and discussion throughout the manuscript.

We have now separated the local scale analysis into a results (3.4) and discussion section (4.4). Figures 7-9 that show the faunal and abiotic changes over time are now located in the

results sections, while Figure 10 that links faunal changes to land use changes remains in the discussion section, as these in our opinion contain a certain level of data interpretation.

4) The authors should consider additional analysis to detail the relationships between biotic and abiotic factors. Currently, a number of comparisons of fauna to abiotic factors (e.g. sediments, nitrate and dissolved oxygen) are made, but there is no clear analysis of these parameters. The conclusions made regarding 'clear links' between for example, DO and groundwater composition are, at this stage, unsubstantiated (eg line 340, 370)

Response: We agree that additional statistical analysis should be presented in the manuscript. Extensive statistical analyses were indeed carried out during the preparation of the manuscript, but as these showed only few significant correlations they were not shown. We now added information on these analyses in the manuscript (section 3.3 and 4.3) and the supplement.

Results: Section 3.3

"Correlation analyses are performed using the Spearman rank correlation coefficient to identify potential relationships between faunal and abiotic parameters, such as well configuration, geology and groundwater quality. In total, 164 parameters are considered, 92 of which are biotic parameters (65 taxonomic parameters and 27 other biotic parameters; see Table S4). Moreover, 72 abiotic parameters from the chemical-physical parameters of the LUBW monitoring are used. Samples are considered significantly correlated if the coefficient is $> 0.5$ or $< -0.5$.

The correlation coefficients reveal that the content of dissolved organic carbon (DOC) of the sampled aquifer water significantly correlates with the amount of biologically usable sediment ($\rho = 0.53$). Both parameters reflect the food content delivered through surface water input. Moreover, the DOC content correlates with the number of individuals and species of the subclass Ostracoda ($\rho = 0.51$; $\rho = 0.53$) and the number of juvenile Copepods ($\rho = 0.51$). Furthermore, there are correlations between the measured depth of the well and the abundance of the subclass Ostracoda and the total number of species ($\rho = 0.51$; $\rho = 0.51$). Also, some statistically significant correlations exist between faunal

 parameters and several anions and cations. These correlations are not further addressed here since only limited data is available for those parameters (mercury, strontium, etc.)."

Discussion: Section 4.3

"The observed correlation between the DOC and Ostracodes can be explained by the fact that the class Ostracoda rely on a temporary surface influence, as they need nutrients (mainly detritus) from the surface and are thus typical for the interstitial and porous aquifers (Mösslacher and Hahn, 2003). A positive relationship between depth and the number of Ostracodes is also observed in Reeves et al. (2007). Moreover, these findings are consistent with a positive relationship between invertebrate density and dissolved and particulate organic matter found in previous studies (Datry et al., 2005; Hahn, 2006; Mösslacher and Notenboom, 1999). Besides the DOC, no correlations are found between faunal parameters and those reflecting food supply for groundwater ecosystems. Thus, food is not a limiting factor for the occurrence of groundwater fauna in the studied wells. In general, only a few significant correlations were found. This is in line with observations by previous studies where correlation analysis revealed significant correlations between chemical variables but no or very few and only weak correlation between abiotic and biotic parameters (Dumas et al., 2001; Griebler et al., 2014b; Hahn, 2006; Koch et al., 2021; Schmidt and Hahn, 2012; Steube et al., 2009). A commonly noted reason for this is that ecosystems are complex multivariate systems exposed to multiple stressors simultaneously (Steube et al., 2009)."

5) Whilst it is understood that there are limitations with the use of historic data in terms of sample collection, there are numerous papers that indicate the issues of using nets to sample groundwater ecosystems without purging wells prior to sampling (including Hahn & Matzke 2005; Sorenson et al 2013; Hancock & Boulton 2009; Korbel et al 2017). The number of species and presence/absence of taxa can possibly be reported with some degree of confidence; however the composition of communities is much harder to describe when wells have not been purged (see https://www.iesc.gov.au/sites/default/files/2024-

05/bioassessment-groundwater-ecosystems-

1.pdf and https://www.iesc.gov.au/sites/default/files/2024-05/bioassessment-groundwater-ecosystems-2.pdf). This is presumably due to specific species habitat & breeding preferences, the artificial well environment as well as the integrity of any well capping. This does not mean that the data cannot be reported as there are some interesting findings, however these issues, at a minimum, need to be discussed in some detail and justifications given for the conclusion made.

Response: We agree that potential limitations of the sampling method should be discussed and added the following sentence in the method section (2.2):

"Limitations regarding the sampling method must be considered when interpreting the faunal results. With the help of a net sampler, only the standing water of a groundwater monitoring well is sampled. This water is affected by filter effects, which can lead to an overrepresentation of larger individuals, such as amphipods (Hahn and Matzke, 2005; Korbel et al., 2017). Larger, more active and sessile organisms, on the other hand, may not be fully captured by sampling using pumps, as they are subject to filter effects, especially in fine sediments. However, studies have shown that the proportion of species and presence/absence of taxa is similar between the two sampling methods (Hahn and Gutjahr, 2014; Hahn and Matzke, 2005; Korbel et al., 2017)."

6) The abiotic conditions are also likely to be impacted by the sampling methodology and lack of purging. Water chemistry is usually studies after wells have been purged (generally a minimum of three times or until water is stable). This is a major concern for this study, the lack of any significant changes abiotic parameters over the year may simply reflect the artificial well environment being sampled rather than being representative of the wider aquifer?

Response: We agree that the abiotic parameters are impacted by the sampling method, but we do not agree that this is a major concern for this study. As written in the last paragraph of the method section, the 144 physico-chemical parameters that were used for the temporal and statistical analysis (correlation analyses, PHATE, etc.) were obtained after pumping and

purging the wells. Also, there is a statistically significant, positive correlation between pumped aquifer samples and standing water samples in terms of electrical conductivity ($\rho = 0.90$), dissolved oxygen content ($\rho = 0.72$), pH value ($\rho = 0.77$), dissolved organic carbon ($\rho = 0.67$) and water temperature ($\rho = 0.68$). Thus, the well parameters seem to reflect the general trend in the aquifer conditions and also in the living conditions of the fauna sampled during the study period.

**Specific comments**

**Introduction**

Comment #1: The main literature is adequately covered in the introduction. There could be more focus given on the importance of long-term data. You could consider making comparisons to long term data sets used for the management of surface aquatic ecosystems, the benefits of such long term studies and how they translate/ inform management directions. This could be incorporated into paragraph starting line 68.

Response: We agree that a comparison to long-term data sets of surface aquatic ecosystems can be useful. Thus, we added the following paragraph:

"Surface waters have been a key focus of aquatic research due to their accessibility and visibility. The assessment of surface waters is typically based on biological, hydro-morphological and physico-chemical criteria and is defined in detail in the European Water Framework Directive (WFD). Accordingly, there is a large number of studies with long-term data on aquatic surface ecosystems. In this context, a recent study collected 1,816 time series from riverine systems in 22 European countries from 1968 to 2020 (714,698 observations) to investigate freshwater biodiversity (Haase et al., 2023). The authors conclude that standardised, long-term and large-scaled monitoring can be used to effectively characterise temporal changes in biodiversity and environmental drivers and identify sites at high risk.

A prime example of long-term monitoring in this field is the Swedish national surface water monitoring program, which began with the first research on the Mälaren lake in 1964 aiming to better understand the eutrophication. This project contributed significantly to understand the effects of climate change, land use and post-glacial rebound on water quality. Today, the combined program comprises monitoring of water chemistry and biodiversity in 114 streams and 110 lakes, and a probability-sampling program includes 4,800 lakes (Fölster et al., 2014). With the advent of the WFD, reference sites from Swedish monitoring were used for inter-calibration of northern continental Europe. Data of this monitoring program 'play a key role in past and present national and international environmental commitments including Swedish environmental objectives, critical load assessments, and many aspects of EU legislation, such as the WFD, Habitats Directive, and Nitrate Vulnerable Zone Directive' (Fölster et al., 2014)."

Comment #2: e.g. line 28-30, English is not correct. Sentence needs re-writing .

Response: We agree and divided the sentence as follows:

"Furthermore, groundwater ecosystems build the largest terrestrial freshwater biome of the world (Griebler et al., 2014a). This habitat is considered species-rich (>100,000 species) with many endemic taxa (Culver and Holsinger, 1992)."

Comment #3: Line 49- issues with formatting are apparent.

Response: We agree and modified the first bracket.

**Methods**

Comment #4: More detail on the groundwater sampling process is required (also bailer not bailor).

Response: We agree and added details (see below).

- Need to state wells are unpurged if this is the case

  Response: In the fourth line of the first paragraph of the method section (2.2) there is a note that standing water is sampled in this step. Further below in the same chapter the sampling of physico-chemical parameters after purging is described.

- How was DO taken, was it using a flow through systems (otherwise please recognise this as a limitation of the design if bailers used).

  Response: The oxygen content of the water is determined in two ways, depending on the origin of the water. The standing water sample is analysed using the HQ40D portable 2-channel multimeter with an Intellical LD0101, luminescence-based/optical probe (see line 142). The pumped aquifer water samples are analysed according to the DIN EN 25814 (in-situ or using a flow-through system). Laboratory measurements are conducted in an accredited laboratory according to DIN EN ISO/IEC 17025.

  We added information on the analysis of the pumped aquifer water in the method section:

  "Aquifer water is analysed in-situ in the field with probes in accordance with the LUBW groundwater sampling guidelines and in line with the applicable DIN standards for electric conductivity (DIN EN 27888), water temperature (DIN 38404 4), pH-value (DIN EN ISO 10523), oxygen (DIN EN 25814, in-situ or using a flow-through system) and base capacity (DIN 38409-7) (Landesanstalt für Umwelt Messungen und Naturschutz Baden-Württemberg, 2013). Laboratory measurements are conducted in an accredited laboratory according to DIN EN ISO/IEC 17025."

- Please specify the exact chemical analysis undertaken (line 127) and remove the 'etc'.

  Response: We agree and added this information.

  "Additional samples were taken to determine the amount of sediment and further chemical analyses (content of dissolved carbon, organic nitrogen, phosphate, ammonium, ortho-phosphate, total phosphate, total bacterial count and total number of cells)."

- Details on how were samples stored/ preserved is required… if this is in supplementary material please refer to this here.

Response: This information is given in the method section:

"The collected samples were stored in a cooling box at about 8 °C until fixation with 96 % ethanol on the same day. The dye Rose Bengal ($C_{20}H_2Cl_4I_4Na_2O_5$ by Thermo Scientific Chemicals) is added until 2015 and Eosin B ($C_{20}H_6Br_2N_2Na_2O_9$) from 2015 onwards, which colours organic matter in a pink hue for easier determination of groundwater fauna."

- What magnification were samples sorted under?

Response: We added this information:

"Faunal samples were sorted on order level and determined on species level under a magnification between 10 and 20."

- Line 138- what is LUBW?

Response: This abbreviation is explained in the first line of the material section: „… the State Office of Environment, Measurements and Nature Conservation (Landesanstalt für Umwelt, Messungen und Naturschutz Baden-Württemberg, LUBW) …"

Comment #5: Line 195. You mention wells dominated by Tardigrades however I cannot find the results for this measure?

Response: The dominance of Tardigrades in one well (Schwäbisch Hall) can be found in Figure 3, showing the faunal community according to taxonomic groups, in light brown.

We added a reference to this fact in the text and also created a data table as a second online supplementary.

**Statistical analysis**

Comment #6: Please provide more detail on the PHATE analysis.

Response: We agree and added more information on the PHATE-analysis (see also Comment#3 Reviewer 1) and how it is applied in this study. Thus, we added the following sentences:

355     "In contrast to most existing methods, a PHATE-analysis allows to obtain biologically interesting structures from increasingly large, modern datasets while accounting for the high level of noise inherent in biological datasets."

"Another advantage of this method is that biological and abiotic parameters can be considered together, allowing identification of further dependency structures."

360     "The selection of parameters is based on previous studies, as well as empirical values on influencing factors from the literature (Hahn, 2006; Koch et al., 2021; Korbel et al., 2018; Stein et al., 2012)."

"From each well, data of all annual measurement campaigns are analysed separately to assess temporal variations in faunal communities and abiotic conditions."

365

Comment #7: The description of stable and unstable conditions needs to be defined better in the text of the manuscript

Response: We agree that stability can be defined in more detail. Therefore, we added information on the categorisation:

370     "The categorisation of measurement sites into stable and unstable according to their condition is based on the variance of faunistic and abiotic (hydrogeochemical) parameters over the period under investigation (Table S3). The variance is parameter-specific and depends on natural and seasonal fluctuations, such as groundwater temperature varying by 1 - 2 °C over the course of a year (Gibert et al., 1994; Taylor and Stefan, 2009).

375     Measurement wells with a total standard deviation over all observed parameters above 14 are classified as unstable wells."

Comment #8: This paper would benefit from a multivariate analysis comparing the fauna to abiotic factors. This would allow investigation into what is causing the changes in biotic communities.

380     e.g.Line 195-200 there is a link between DO and decreased abundance there is a need for some statistical analysis here, maybe on a broader scale looking at DO and fauna abundances and richness within regions and over years. Lines 235-240 indicate that sediment is responsible for some trends,

but I cannot find the statistical analysis that indicates this relationship. This section is a little vague and needs further analysis. This paragraph also lacks references to figures.

385      Response: We agree that a multivariate analysis can be helpful to explain changes in communities. For this reason, a PHATE analysis and correlation analyses were carried out as part of this study.

However, we do not agree that further statistical analyses on the relationship between oxygen, abundance and diversity are reasonable here. With a few exceptions (e.g. in Zienken), all

390      monitoring sites had DO concentrations of $> 1$ mg/l during the study period, which is considered the minimum for fauna colonisation (Griebler et al., 2014b). Fluctuating oxygen contents above 1 mg/l are therefore not expected to influence groundwater fauna in the study area. For this reason, oxygen contents are not used in the statistical evaluation method (PHATE-analysis) in order to avoid spurious correlations.

395      The correlation analysis does not indicate any relationship between the amount of sediment and other parameters, except the proportion of biologically usable sediment. The above-mentioned trend in sediment content in the PHATE-analysis is due to very local effects over time in conjunction with other fluctuating parameters, as stated in the first line in section 4.3. Nevertheless, we added the following information on this issue.

400      "Although, the correlation analysis does not indicate any relationship between the amount of sediment and other parameters, except the proportion of biologically usable sediment, a trend in sediment content in the PHATE-analysis is visible. This trend is due to very local effects over time in conjunction with other fluctuating parameters."

The vague character of this paragraph is based on the fact that no more precise statements

405      and evidence can be made based on the limited amount of data (16 wells with 282 samplings over 20 years) and the general complexity of this habitat.

**Faunistic overview (line 165 onwards)**

Comment #9: Total abundance can also be influenced by the time between sampling- has this been considered in the analysis.

410     Response: We agree that the time between individual samplings can also influence the
        abundance. Thus, we added this information as follows:

              "Potential reasons for this are a high hydrogeological and hydro-chemical heterogeneity
              in combination with a small number of monitoring wells, the time between sampling,
              superimposing effects of local influences and site-specific parameters linked to variations
415           in topography and geology."

Comment #10: The increased abundance of Tardigrades within wells, may simply indicate
contamination from the surface (as stated e.g. moss or terrestrial carbon contamination within the
well)- rather than a change in the broader aquifer ecosystem- such factor should be included in the
420   discussion section.

        Response: We do not think that the well is contaminated in a manner as described above.
        This measurement well is located on a former bombing range, which now consists of a large
        and dense wooded area. The well itself is only about six metres deep and is located between
        two (currently dried-out) lakes. Since the well appears to be intact and no anomalies were
425     found during sampling, we think it is more likely that there is an influence from surface water
        related to the dried-out lakes in this area. However, we added a statement about the possibility
        of contamination in the discussion section:

              "For example, in Schwäbisch Hall, where the well appears to be intact and no anomalies
              were found during sampling, it is more likely that changes in the faunal community and
430           abundance are related to the dried-out lakes in the wooded area, but potential
              contaminations cannot be ruled out."

**Section 4.3**

Comment #11: This appears to be results rather than discussion. The discussion here is likely to be
very relevant to the paper, but would be improved by a total restructure of the paper.

435    Response: We agree that the separation into results and discussion sections is currently not ideal and should be improved. We have therefore reviewed and rewritten the content of these sections.

**Section 4.4**

Comment #12: This section needs to be re-written removing the claims not backed by statistical
440    results. It is suggested that the majority of this is moved to the results section, and a new section in the discussion is written addressing the aims of the manuscript and the findings of the results. This section should be restructured to engage more with relevant literature describing the results, rather than presenting results.

    Response: As mentioned before (Major Comment#3) we have restructured this section
445    according to the suggestions made above.

Comment #13: Figure 5: **Electrical** Conductivity

    Response: We agree and replaced "electric" with "electrical" throughout the manuscript and supplement.
450

Comment #14: Figure 6: what is meant by faunistic and hydrochemical stable/ unstable- more explanation is required

    Response: We agree that more explanation is necessary. Thus, we added more information in the text (see also Comment#7).
455

Comment #15: Figures 7-9 are confusing. What does the picture (presumably oligochaete) with a (1) mean in Fig 7, or the cyclopoida in 2017 with a (6) in fig 7.  [ same issues in Figure 9 & 10]. A list of stygophilic/ stygobiont species could be presented/ indicated in current Figures 7-10.

    Response: The sketches of the animals represent selected species ("key species") of the
460    measurement wells. The sketches are always close to the bar of their (usually first) occurrence. The small number in brackets refers to the corresponding legend item (name, colour).

We agree that information on the classification of the species into stygobiont or non-stygobiont could be helpful here. Thus, stygobiont species have now been marked with a superscript cross symbol along with the species name in the legends of the figures.

465